# CRISPR-Cas effector specificity and cleavage site determine phage escape outcomes

**Michael A. Schelling, Giang T. Nguyen, Dipali G. Sashital** *

Roy J. Carver Department of Biochemistry, Biophysics and Molecular Biology, Iowa State University, Ames, Iowa, United States of America

* sashital@iastate.edu

## Abstract

CRISPR-mediated interference relies on complementarity between a guiding CRISPR RNA (crRNA) and target nucleic acids to provide defense against bacteriophage. Phages escape CRISPR-based immunity mainly through mutations in the protospacer adjacent motif (PAM) and seed regions. However, previous specificity studies of Cas effectors, including the class 2 endonuclease Cas12a, have revealed a high degree of tolerance of single mismatches. The effect of this mismatch tolerance has not been extensively studied in the context of phage defense. Here, we tested defense against lambda phage provided by Cas12a-crRNAs containing preexisting mismatches against the genomic targets in phage DNA. We find that most preexisting crRNA mismatches lead to phage escape, regardless of whether the mismatches ablate Cas12a cleavage in vitro. We used high-throughput sequencing to examine the target regions of phage genomes following CRISPR challenge. Mismatches at all locations in the target accelerated emergence of mutant phage, including mismatches that greatly slowed cleavage in vitro. Unexpectedly, our results reveal that a preexisting mismatch in the PAM-distal region results in selection of mutations in the PAM-distal region of the target. In vitro cleavage and phage competition assays show that dual PAM-distal mismatches are significantly more deleterious than combinations of seed and PAM-distal mismatches, resulting in this selection. However, similar experiments with Cas9 did not result in emergence of PAM-distal mismatches, suggesting that cut-site location and subsequent DNA repair may influence the location of escape mutations within target regions. Expression of multiple mismatched crRNAs prevented new mutations from arising in multiple targeted locations, allowing Cas12a mismatch tolerance to provide stronger and longer-term protection. These results demonstrate that Cas effector mismatch tolerance, existing target mismatches, and cleavage site strongly influence phage evolution.

**Data Availability Statement:** All relevant data are within the paper and its Supporting Information files." Scripts used to analyze high-throughput sequencing data are available at https://github.com/alexsq2/lambda-phage-CRISPR-mutants.

## Introduction

CRISPR-Cas (clustered regularly interspaced short palindromic repeats-CRISPR associated) systems are adaptive immune systems found in bacteria and archaea [1–3]. These systems use ribonucleoprotein effector complexes to find and destroy foreign nucleic acids that have entered the cell. CRISPR effector complexes are guided by a CRISPR RNA (crRNA) to a

**Funding:** Financial support for this research was provided by National Science Foundation award 1652661 (to D.G.S.). The funders had no role in study design, data collection and analysis, decision to publish, or preparation of the manuscript.

**Competing interests:** The authors declare no competing interests.

nucleic acid target that is complementary to a section of the crRNA called the spacer. Bacteria can acquire new spacer sequences that allow them to mount an immune response against threats they have not previously encountered [4,5].

An important function of CRISPR-Cas systems is to prevent infection by bacteriophages, which can have significant impact on the composition of a bacterial population [6–9]. As a CRISPR effector complex requires a match between its crRNA and a target to engage in interference, selection occurs for phages with mutations in targeted genomic regions [10–12]. Mutations in CRISPR targets create mismatches between the target and the crRNA that weaken the base-pairing interaction [13–15], slowing or stopping target matching by Cas effectors [16] and allowing phages to safely multiply in the bacterial cell. Different CRISPR-Cas systems have DNA or RNA as a primary target and prevent infection at the cellular and population level [17–22]. Target binding is more stringent in DNA targeting systems, mitigating highly damaging off-target cleavage of host DNA [23]. In these systems, a protospacer adjacent motif (PAM) next to the target is required to initiate base pairing [24–27]. Complete base pairing is especially important in the region next to the PAM, called the seed region [28–34]. Accordingly, mutations that allow phages to escape CRISPR immunity are often single mutations in the PAM or seed region [10–12,35].

There have been multiple proposed but noncompeting mechanisms for this mutagenesis. Mutants may exist due to natural genetic variation in the population and these could be selected through CRISPR pressure and become dominant in the population over time [11,36]. Alternatively, escape mutations may be generated by Cas effector cleavage and subsequent error prone DNA repair [37]. It has been shown that cleavage by Cas effectors causes large deletions to appear in the genome of T4 phage, resulting in loss of the crRNA target sequence [38]. Lambda phage encoded Red recombinase has been implicated in generating mutations in CRISPR targets that allow escape [39]. It remains unclear to what degree each of these mutagenesis pathways contribute to phage escape under different conditions.

Escape mutations are evident in natural settings as bacterial CRISPRs often contain mismatched spacers to common mobile genetic elements and the genomes of phages [27,40–43]. New spacers are added at the leader end of CRISPR arrays and these new spacers are more likely to match a target in a phage genome exactly [44,45]. Genomic evidence also shows that spacer sequences in a CRISPR array do not commonly develop mutations and are fixed once they are acquired [12,46]. Instead, spacers are lost from the array entirely when they lose effectiveness as mutations accumulate in targeted genomic elements.

Mismatched spacers may provide some benefit to the host. Spacers against mutated targets drive some Cas effectors towards primed spacer acquisition, in which new spacers are preferentially acquired from genomes targeted by the Cas effector [11,31,47–50]. Mismatched crRNAs may also provide low-level immunity through continued target cleavage. Cas effectors tolerate mismatches between the crRNA and target, allowing cleavage of mutated targets [26,28,29,51–55]. This lax specificity may partially prevent phage escape. The type V-A Cas12a effector has been shown to tolerate multiple mismatches between its guiding crRNA and the target in vitro [53,56,57] leading to rare off-target genome edits in cells [57–61]. However, this mismatch tolerance varies depending on the crRNA sequence and type of mismatch. Multiple Cas12a variants also have the ability to nick double-stranded DNA targets with many (3–4) mismatches [53,62]. These in vitro observations raise the question of how the specificity of Cas12a affects its role in preventing infection by phage with target mutations.

To test this, we subjected bacteria expressing Cas12a and crRNAs with varying target mismatches to phage infection. We found unexpected discrepancies between the effect of crRNA mismatches on target cleavage in vitro and survival of bacteria upon phage infection. This led

us to monitor mutant emergence in phage populations. Using high-throughput sequencing, we discovered enrichment of a large variety of mutations when the phage was targeted by different crRNAs with and without mismatches. Single crRNA mismatches, even those outside of the seed region, had a drastic effect on the ability of bacteria to survive phage exposure, demonstrating the importance of spacer diversity as mutations in target genomic regions propagate. crRNA mismatches increased the rate at which mutant phage arose in the population. Mismatches in the PAM-distal region led to mutations in proximity to the original mismatch, leading to highly deleterious combinations of PAM-distal mismatches. Although similar mismatches were also deleterious for Cas9 cleavage, similar mutants did not emerge when phage was challenged with Cas9-crRNA complexes bearing PAM-distal mutations, suggesting that PAM-distal cleavage by Cas12a may result in more phage escape via PAM-distal mutations. Overall, we find that phage populations evolve in different ways to resist CRISPR interference depending on Cas effector specificity, existing crRNA-target mismatches, the location of CRISPR targets in the phage genome, and the cleavage site of the Cas effector.

## Results

### crRNA mismatches throughout the spacer decrease phage protection provided by Cas12a

To investigate the effect of crRNA mismatches on phage immunity provided by Cas12a, we developed a heterologous type V-A CRISPR-Cas12a system in *Escherichia coli*. We expressed Cas12a from *Francisella novicida* and various pre-crRNAs from 2 different plasmids in *E. coli* K12 using a strong inducible promoter ($P_{BAD}$) or a relatively weak constitutive promoter [49,63]. We infected these cells with lambda phage to measure the immunity provided by Cas12a (Fig 1A). We used $\lambda_{vir}$, a mutant of lambda phage that cannot engage in lysogeny and is locked into the lytic mode of replication. This eliminates CRISPR self-targeting that could occur if a target phage becomes a lysogen in the bacterial genome. We chose 2 lambda genomic targets: one target was in an intergenic region upstream of gene J and the other target was inside the coding region of gene L (Fig 1A). Both genes encode essential structural tail tip proteins. The Cas12a expression system exhibited a high level of protection for both promoters, with targeting crRNAs showing about $10^6$ fold less phage infection than the non-targeting control (Fig 1B).

We designed 4 mutant crRNAs with varying levels of in vitro cleavage defects (Fig 1C) and tested their effects on phage defense (Fig 1B). These mismatches spanned the target with 1 in the seed region, 1 in the mid-target region, and 2 in the PAM-distal region. We observed a strong defect for the seed mutant when we used the weaker promoter to express Cas12a. However, this defect was reduced upon Cas12a overexpression using the stronger promoter (Fig 1B), consistent with the defect being caused by reduced Cas12a targeting. Mid-target and PAM-distal mismatches caused almost no visible defects in protection for the gene J target and small defects for the gene L target when Cas12a expression was controlled by the stronger promoter. These results correlated with the cleavage defects measured in vitro for the corresponding mismatched crRNAs, where seed mismatches had stronger defects than other mismatches but still enabled complete cleavage (Figs 1C and S1 and S1 Data), consistent with previous results [53,54]. However, when Cas12a expression was controlled by the weaker promoter, we observed a large loss of protection for the mid-target mismatched cRNA targeting gene J, which had no significant loss of cleavage in our in vitro assay (Figs 1B, 1C, and S1 and S1 Data). These results suggest that factors outside of reduced targeting may affect Cas12a-mediated protection at low expression levels.

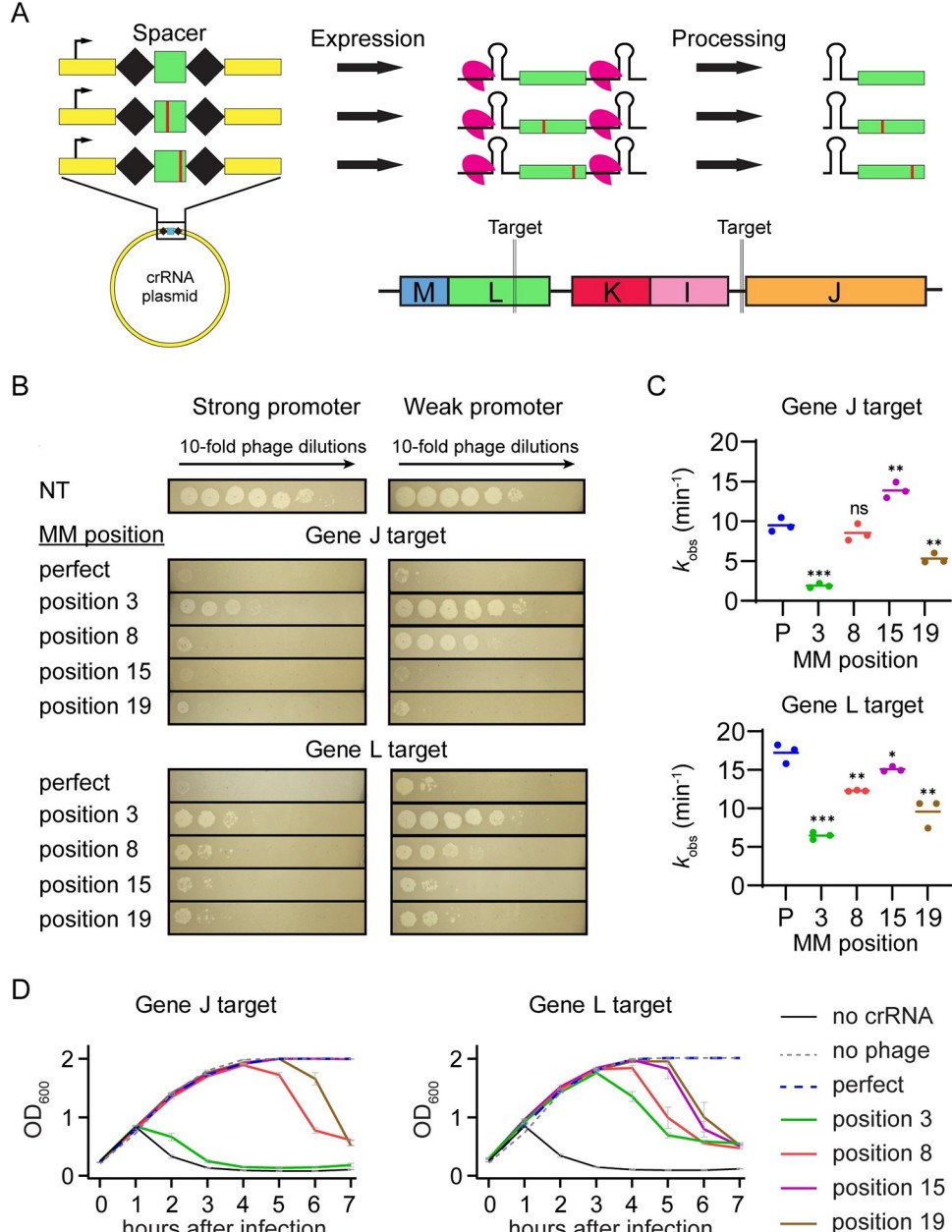

**Fig 1. Effects of mismatched crRNAs on Cas12a-mediated phage defense.** (A) Schematic of crRNA expression and processing by FnCas12a and crRNA phage target locations. crRNA mismatches were introduced by mutating individual nucleotides in the spacer sequence. After expression of the pre-crRNA, Cas12a processes it into a guiding crRNA that partially matches the lambda phage genome targets upstream of gene J and in the coding region of gene L. See S1a for target and crRNA spacer sequences. (B) Measurement of phage protection provided by crRNAs with and without target mismatches. Spot assays were performed with bacteria expressing FnCas12a and a crRNA construct that either perfectly matches the lambda phage genome (perfect) or has a crRNA mismatch (MM) at a position in the spacer (position x, sequences shown in S1A Fig). A non-targeting crRNA construct (NT) was used as a negative control. Lambda phage was spotted on cells with 10-fold decreasing concentration at each spot going from left to right. Expression of FnCas12a and pre-crRNAs were controlled by a stronger inducible PBAD promoter or a weaker constitutive promoter. (C) Observed rate constants for in vitro cleavage by Cas12a armed with crRNAs containing target mismatches. Plasmids bearing target sequences for gene J or L were used to measure Cas12a cleavage. Mismatch positions or perfect crRNAs (P) are indicated on the horizontal axis. Data from 3 replicates are plotted. * $P \leq 0.05$, ** $P \leq 0.01$, *** $P \leq 0.001$, ns = no significant difference compared to the perfect crRNA based on unpaired two-tailed $t$ test. See S1B and S1C Fig and S1 Data for gels and quantification. See S1 Fig for crRNA and target sequences, representative gels, and fit data. (D) Growth curves for *E. coli* expressing mismatched crRNAs following phage

infection. Bacteria containing the $P_{BAD}$ FnCas12a expression plasmid and various crRNA expression plasmids were inoculated in liquid culture and induced immediately. Lambda phage was added 1.5 h after inoculation and $OD_{600}$ measurements were taken every hour. Bacteria expressed no cRNA, a crRNA with no mismatches to the target (perfect) or a crRNA with a mismatch at the indicated position (position x). A no phage condition was performed as a negative control. The average of 2 replicates is plotted, with error bars representing standard deviation. See S2 Data for quantified data.

We then tested the effects of mismatched crRNAs in liquid culture when Cas12a was expressed from the stronger promoter. Cells containing a matching crRNA grew at the same rate as cells that were uninfected with phage, demonstrating complete Cas12a protection in the time frame tested (Fig 1D and S2 Data). In contrast, most mismatched crRNAs caused lysis to occur regardless of the mismatch location in the target. For the gene J target, a crRNA mismatch in the seed region caused lysis to begin 1 h after infection, similar to a culture bearing a non-targeting crRNA. This indicated that the seed mismatch was allowing nearly full phage escape, consistent with this mismatch causing the largest reduction of target cleavage in vitro (Fig 1C).

In contrast, the seed mismatched crRNA against gene L provided protection for several hours post infection, with lysis beginning 3 h post-infection (Fig 1D). Mismatches in the mid- or PAM-distal region offered protection until 4 or 5 hours following infection. Interestingly, the rate of cleavage for these crRNAs did not always correlate with the level of protection provided in liquid culture (Fig 1C and 1D). Some crRNA mismatches that caused small decreases or no significant effect on cleavage rates in vitro led to lysis of the culture (e.g., gene J position 8 and gene L position 15). These results indicate that loss of cleavage caused by crRNA mismatches did not completely account for loss of immunity.

## Phage target mutations depend on location of existing mismatches

Our initial results showed that crRNA mismatches have less of an effect on solid media than in liquid culture when Cas12a is expressed from a strong promoter. We hypothesized that these differences were caused by phage mutation upon CRISPR immune pressure. Unlike on solid medium, phage mutants that arise can quickly and uniformly spread throughout the culture in a liquid medium. Thus, preexisting mismatches or mismatches that arise through imperfect DNA repair following Cas12a cleavage may accelerate the selection for escape mutants as they quickly spread throughout the population, causing lysis in liquid culture. These results overall suggested that loss of protection from crRNA mismatches is due in part to emergence of phage mutants that further disable CRISPR interference.

To test this hypothesis, we investigated mutations that arose in phage populations in response to CRISPR pressure by Cas12a effector complexes with or without preexisting crRNA mismatches (Fig 2A). We isolated phage from liquid cultures of *E. coli* expressing matching or mismatched crRNAs and PCR amplified the regions of the phage genome that were being targeted. High-throughput sequencing was then used to identify mutations in the target regions (S3 Data).

We first quantified the percent of the phage population that had mutations in genomic regions targeted by Cas12a over time in liquid culture. We observed mutations within the targeted region of PCR amplicon sequences, but not outside of the target region. Phages targeted by a matching crRNA gradually developed target mutations over time, with about 20% of the phage population becoming mutated after the 8 h time course (Fig 2B and S3 Data). A crRNA mismatch at any of the positions we tested led to a large acceleration of mutant emergence causing the phage population to become almost entirely mutated after 4 h. Interestingly, phages exposed to bacteria expressing crRNAs with a seed mismatch also rapidly mutated,

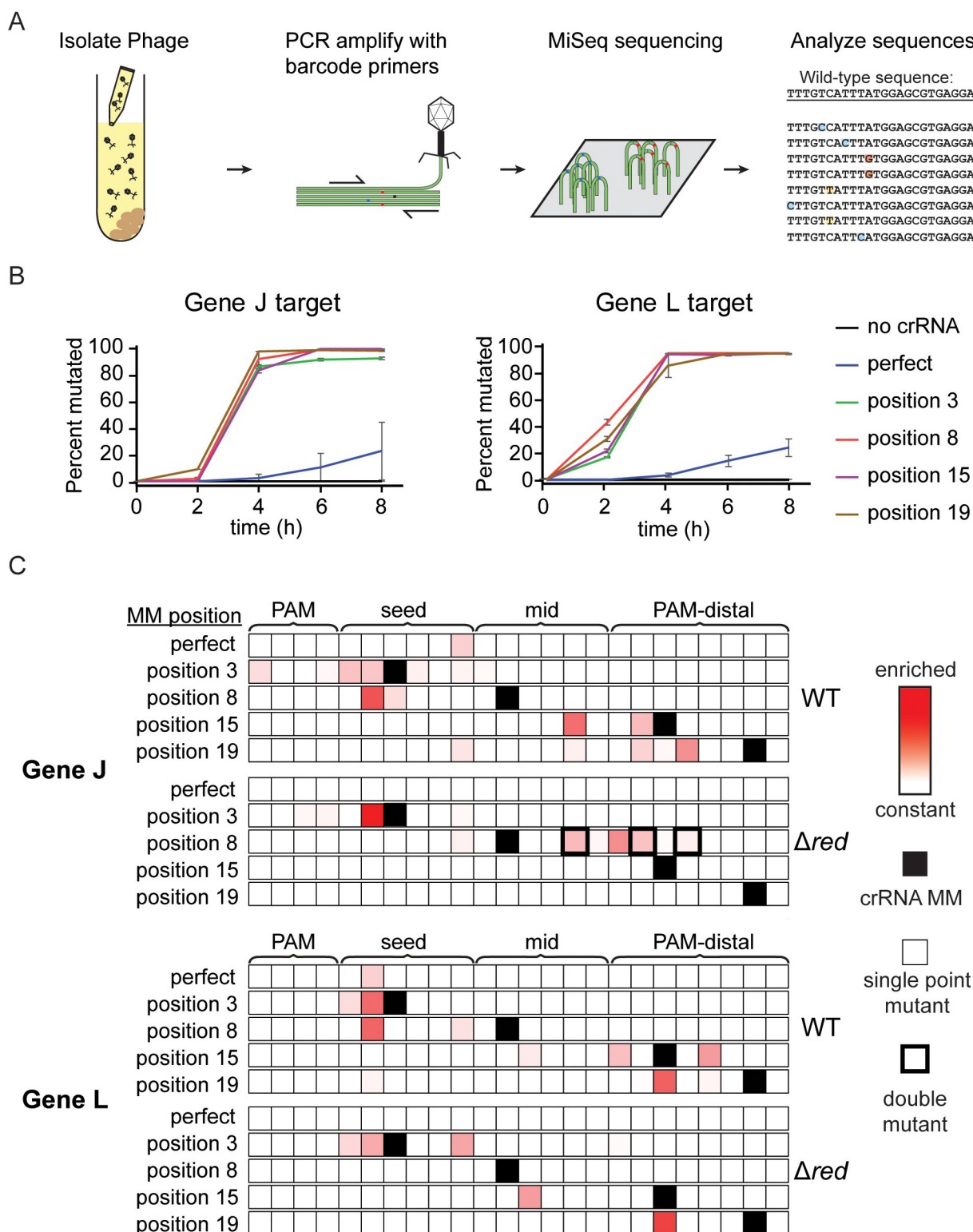

**Fig 2. crRNA mismatches cause emergence of diverse lambda phage mutations.** (A) Schematic of workflow for determining the genetic diversity of phage exposed to interference by Cas12a. Phage samples were collected from liquid cultures at various time points and the target region was PCR amplified. Mutations were observed using MiSeq high-throughput sequencing of these amplicons. (B) Line graph tracking the fraction of phage with mutated target sequences over time. Samples were taken from liquid cultures at time points after phage infection. The "0 h" samples were taken directly after addition of phage to the culture tubes. The average of 2 replicates are plotted with error bars representing

standard deviation. See S3 Data for quantified data. (C) Heat maps showing the location of enriched phage mutations in target regions at the 8 h time point for gene J or L targets. Z-scores for abundance of single-nucleotide variants, including nucleotide identity changes or deletions, were determined for each sample relative to the non-targeted control phage population. Experiments were performed using $\lambda_{vir}$ phage (WT) and $\lambda_{vir}$ phage with the *red* operon deleted (Δ*red*). Enriched sequences indicate high Z-scores. Z-scores range from 0 (white) to 10.3 (darkest red). Single-nucleotide deletions are shown at adjacent position to the 3′ side. Positions with crRNA mismatches are labeled with solid black boxes in the heat map. A thin outline indicates that the majority of sequences contain single point mutations at these positions while a thick outline indicates that the majority of sequences contain multiple point mutations at these positions. See S3 Data for quantification of variant abundance.

even though our in vitro results showed the original crRNA mismatches were highly deleterious for target cleavage (Fig 1C). For most individual replicates of our samples, we did not observe substantial variability in the distribution of mutations after the phage population became highly mutated (S2A Fig and S3 Data). We therefore chose to focus on the longest time point (8 h) for further analysis.

As previously shown in other CRISPR systems [10,35,64], phage populations targeted with a perfectly matching crRNA developed mutations in the seed region of the target (Fig 2C and S3 Data). When the sequences of the crRNA plasmids were changed to create mismatches between the crRNA and the phage genome target, the position of individual point mutations within the phage target became substantially more variable. A crRNA mismatch at position 3 for the crRNA targeting the region upstream of gene J caused 9 different individual point mutations to appear at 8 positions spread across the PAM and seed, none at position 3 as expected given the preexisting mismatch (Fig 2C). For the gene L target, a crRNA mismatch at position 3 only caused 2 different mutations to appear, with one of them being the predominant mutation seen when targeting with a matching crRNA. Mismatches in the mid-target region at position 8 also caused seed mutations to arise. Surprisingly, PAM-distal crRNA mismatches caused enrichment of PAM-distal mutations and prevented nearly all seed mutations from emerging. Most of the mutations present in liquid culture were also observed when sequencing phage from spot assays, although the distribution differed in some cases (S2B Fig and S4 Data). This indicates that the differences we observed between our solid media and liquid cultures experiments were caused by the increased mobility of phages in liquid culture and were unrelated to the types and location of mutations that could arise.

The genomic context of target sequences had a clear effect on the types of mutants that emerged (S2C Fig and S3 Data). For the matching crRNA targeting the region upstream of gene J, the most common mutation observed was a single-nucleotide deletion at position 6. The most common mutation for the gene L target was a single nucleotide substitution at position 2 which is a wobble base position in the codon. Similar to the other mismatched crRNA constructs targeting gene L, most mutations we observed were either silent or caused amino acid changes from valine, threonine or serine to alanine or from proline to leucine. No deletions were observed in the gene L target in any samples with crRNA mismatches, while deletions were observed in the gene J upstream target in samples with crRNA mismatches at positions 15 and 19. This difference in mutational variability reflects the more vulnerable target region of gene L where base substitutions are likely to change the amino acid sequence of the protein and single deletions will cause frame-shifts. The relatively weak constraints on viable mutations in the upstream region of gene J may enable more routes for escape from Cas12a targeting, resulting in the loss of protection at earlier time points (Fig 1D).

## Phage mutations can arise following exposure to Cas12a

Our results indicate that mutations can arise rapidly in regions targeted by Cas12a when a preexisting mismatch is present between the crRNA and target. It is possible that Cas12a targeting selects mutant phages that are present in the population at the time of infection. We

investigated whether the mutated phage we observed in our CRISPR active samples were present in negative control samples. While many of the single-nucleotide substitutions that were enriched following Cas12a targeting were present at very low levels in the control sample, we could not distinguish actual nucleotide variations from sequencing or PCR error (S3 Fig and S3 Data). However, we did observe that 2 out of 3 sequences containing single-nucleotide deletions that were consistently present in control samples were the only 2 deletion mutants that became highly enriched in the escaped mutant phage population (S2C and S4 Figs and S3 Data). These results suggest that deletion mutants that were enriched upon Cas12a-mediated selection were preexisting in the population.

To further test this, we designed crRNAs targeting non-essential regions in the lambda phage genome (S5 Fig and S5 Data). Cas12a-mediated defense against lambda phage using these crRNAs caused large deletions to appear based on recombination at microhomology sites, as has been previously observed [38] (S5A and S5B Fig). We used long-read sequencing to determine whether these regions of the wild-type lambda phage population contained the same deletions. Similar to the single-nucleotide deletions, microhomology-mediated deletions that were enriched upon Cas12a-mediated selection were among the most abundant mutations preexisting in the wild-type population (S5C Fig and S5 Data). Together, these results suggest that mutants that emerge upon Cas12a targeting may be selected from natural genetic variants in the population.

Our observation that enriched deletion mutations preexist in the wild-type population does not rule out the possibility that mutations may be actively acquired following Cas12a cleavage. Previous studies have suggested that DNA cleavage by Cas9 or Cas12a may induce mutation in the target site through DNA repair [35,38]. To test this possibility, we deleted the *red* operon from the $\lambda_{vir}$ genome, which has previously been implicated in escape mutation enrichment following targeting by Cas9 [39]. We performed liquid culture infection assays using the Δ*red* phage in cultures expressing Cas12a and panel of crRNAs described above (S6A Fig and S2 Data). Unlike with wild-type phage, infection with the Δ*red* phage did not cause lysis in most cultures expressing a perfect or mismatched crRNA.

We next sequenced phage populations harvested from cultures 8 h after infection (Figs 2C and S6B and S3 Data). Consistent with the lack of lysis in many cultures, we did not observe mutants arising in cultures expressing either perfect crRNA, the PAM-distal mismatched crRNAs targeting gene J, nor the mid-target mismatched crRNA targeting gene L. For most of the remaining crRNAs, we observed mutations that were observed in the wild-type phage populations challenged with the same crRNA, although the number of mutants and distribution of these mutants varied between phage strains.

One notable exception where we observed new mutations arising in the Δ*red* strain was for the phage challenged with the mid-target (position 8) mismatched crRNA targeting gene J (Fig 2C and S3 Data). Surprisingly, we observed a range of double mutations in the mid and PAM-distal region in Δ*red* phage challenged with this crRNA (Figs 2C, S6B, and S6C and S3 Data). Importantly, all double mutants observed in individual replicates contained unique mutations and appeared to originate with a single point mutation (S6C Fig). These results strongly suggest an active mechanism of mutant generation unrelated to the *red* operon. Overall, our results provide evidence that both preexisting and actively acquired mutations may be selected during Cas12a-mediated immunity.

## Multiple mismatches in the PAM-distal region allow phage escape from Cas12a

A striking result from our sequencing of mutant phage populations was the emergence of PAM-distal mutants upon challenge with crRNAs containing PAM-distal mismatches. Given

that seed mutants appeared when other Cas12a crRNAs were used, these results suggested that multiple PAM-distal mismatches are at least as deleterious for Cas12a cleavage as a seed mismatch combined with a PAM-distal mismatch. It has been shown that pairs of mismatches are more deleterious for Cas12a cleavage when they are closer together, including when both are in the PAM-distal region [53,56]. Together, these results suggest that double mismatches in the PAM-distal region can lead to phage escape from Cas12a.

To test this, we added second PAM-distal crRNA mismatches to crRNAs targeting gene J that initially contained a single PAM-distal mismatch. We chose the second mismatch position based on phage mutants that appeared when exposed to the original mismatched crRNA (Figs 2C and S7A). Adding a second mismatch at position 14 to the crRNA that contained a mismatch at position 15 caused a small defect in phage protection (Fig 3A). Although this mismatch pair had small effects on the overall cleavage rate in vitro regardless of the mismatch type at position 14 (Figs 3B and S7B–S7C and S1 Data), we did observe a cleavage defect, in which the DNA was nicked by Cas12a through cleavage of only 1 strand (S7B Fig). This defect in second-strand cleavage may allow more phage infection, resulting in partial loss of phage defense on solid media (Fig 3A). Consistently, bacteria expressing a crRNA with a position 15 mismatch did not lyse in liquid culture (Fig 1D), despite the emergence of the position 14 mutation (Fig 2C). In contrast, adding any type of second mismatch at position 16 to a crRNA that contained the original mismatch at position 19 caused nearly complete loss of protection (Fig 3A) consistent with lysis observed for the position 19 mismatched crRNA in liquid culture (Fig 1D) and the dramatic loss of cleavage activity in vitro (Figs 3B and S7B–S7C and S1 Data).

We next investigated why PAM-distal mutations may be preferentially selected over PAM or seed mutations that may be highly deleterious to Cas12a cleavage on their own. Notably, while PAM and seed mutations were not highly enriched for wild-type phage challenged with PAM-distal mismatched crRNAs in liquid culture (Fig 2C), we did observe PAM and seed mutants when we assayed the phage population present in spot assays on solid media (S2B Fig). This difference may be due to competition between different mutant phages, in which phages bearing mutations that are more deleterious to Cas12a cleavage may outcompete less deleterious mutants. Such competition is more likely to occur in liquid media where phages are mobile.

To test this hypothesis, we isolated 2 mutant phages selected upon targeting with the position 15 mismatched crRNA (MM15) targeting gene L (see Methods). The mutant phages contained a single point mutation in either the seed (A2T) or the PAM-distal region (G17T) of the gene L target region. As expected, the seed mutant phage caused a far greater loss of protection than the PAM-distal mutant when the 2 mutant phages were used to infect bacteria expressing Cas12a and the perfectly matching crRNA in phage spotting assays (Fig 3C). In contrast, both mutants caused a similar loss of protection in cells expressing the MM15 crRNA.

We next tested the extent to which these target mutations cause Cas12a cleavage defects using both the perfect crRNA and the MM15 crRNA (Figs 3D and S8 and S1 Data). Cleavage of the A2T seed mutant target was reduced by 15 to 20× in comparison to the WT target, and we did not observe a significant difference in cleavage of this target by Cas12a bearing either the perfect or MM15 crRNA (Fig 3D). Strikingly, Cas12a cleavage was reduced by approximately 90× when the G17T target was cleaved with Cas12a bearing the MM15 crRNA. Importantly, this loss of cleavage upon introduction of 2 PAM-distal mismatches was significantly more deleterious than the cleavage defect observed when seed and PAM-distal mismatches were combined (MM15/A2T combination).

Together, our results strongly suggest that PAM-distal mutants emerge upon challenge with crRNAs bearing PAM-distal mismatches because 2 PAM-distal mismatches can be more deleterious to Cas12a cleavage than a seed and a PAM-distal mismatch. To directly test this,

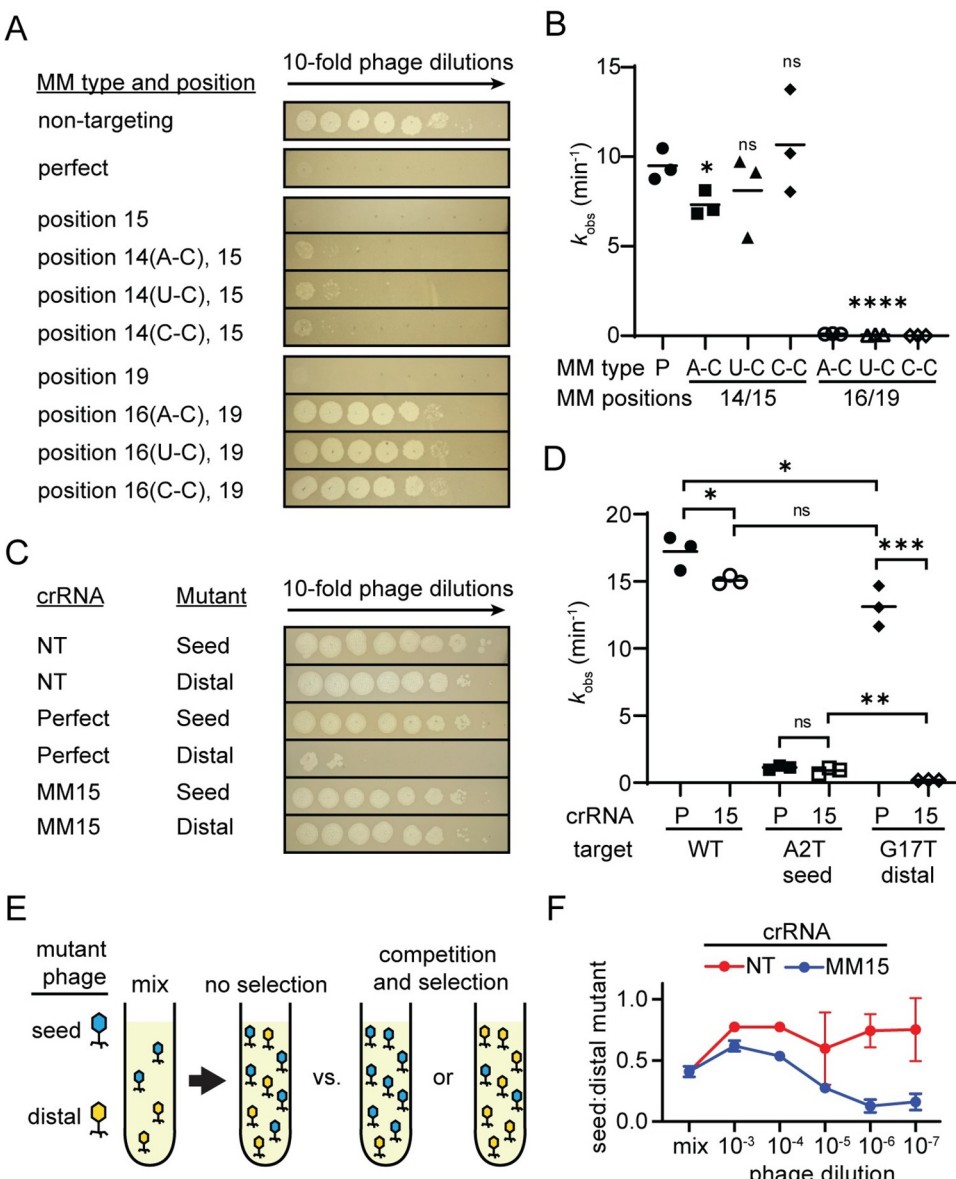

**Fig 3. Two PAM-distal mismatches are more deleterious than seed mismatches.** (A) Spot assays performed using *E. coli* expressing FnCas12a and a crRNA that perfectly matches the lambda phage gene J target (perfect) or has mismatches at the indicated positions. Three types of second mismatches were added and the type of the mismatch is indicated in parenthesis next to the position. See S7A Fig for crRNA spacer sequences. (B) Observed rate constants for in vitro cleavage by Cas12a armed with crRNAs containing 2 target mismatches. Cleavage was measured for plasmid DNA containing a gene J target. The types of mismatches for the second mismatch are indicated. * $P \leq 0.05$, **** $P \leq 0.0001$, ns $P > 0.05$ compared to the perfectly matching crRNA based on unpaired two-tailed *t* test. See S7B and S7C Fig and S1 Data for gels, and quantified and fit data. (C) Phage spot assays for target mutant phages isolated upon challenge with Cas12a programmed with a position 15 mismatched crRNA targeting gene L. Spot assays were performed with *E. coli* expressing a non-targeting crRNA (NT), a crRNA that perfectly matched wild-type phage (Perfect), or the crRNA with a mismatch at position 15 (MM15). Phage mutations were in the seed (A2T) or PAM-distal (G17T) region. (D) Observed rate constants for in vitro Cas12a cleavage of plasmids bearing wild-type (WT), seed mutant (A2T), or PAM-distal mutant (G17T) gene L target sequences. Cas12a cleavage was measured for both the perfectly matched crRNA (P) or the MM15 crRNA (15). Significance was tested pairwise for all crRNA/target combination by unpaired two-tailed *t* test. * $P \leq 0.05$, ** $P \leq 0.01$, *** $P \leq 0.001$, ns $P > 0.05$. Pairwise comparisons for which $P$ value are not indicated had a $P < 0.0001$. See S8 Fig and S1 Data for crRNA and target sequences, gels, and quantified and fit data. (E) Schematic of competition assay. Two mutant phages, A2T and G17T, were mixed at approximately equal titers. *E. coli* expressing Cas12a and the position 15 mismatched crRNA were infected with a dilution series of the mutant phage mix. Lysates were harvested and the proportion of each mutant phage was

determined by high-throughput sequencing. (F) Ratio of seed mutant (A2T) to PAM-distal mutant (G17T) following lysis of cultures infected with a dilution series of the mixed phage. *E. coli* expressed Cas12a and either a non-targeting (NT, red) or position 15 mismatched (MM15, blue) crRNA. "Mix" indicates the initial mixture of phage. The average of 3 replicates is shown, with error bars indicating standard deviation. See S6 Data for raw and quantified data.

we performed a competition assay in liquid culture in which cells expressing either a non-targeting or the MM15 crRNA were coinfected with a mixture of the A2T and G17T mutant phages (Fig 3E and 3F). We performed the competition at decreasing phage concentrations, allowing competition to occur as the phages were propagated in the culture. Lysates were sampled after 8 h, the target region was PCR amplified, and the abundance of each mutant was determined by high-throughput sequencing (S6 Data).

Notably, although both the A2T and G17T substitutions are silent mutations, the A2T mutant slightly outcompeted the G17T mutant in cultures expressing the non-targeting crRNA (Fig 3F), suggesting that the A2T mutant phage may be slightly more fit than the G17T mutant. In contrast, when coinfection was performed in cultures expressing the MM15 crRNA, the G17T mutant became dominant in the phage population when cultures were infected with highly diluted phage mixtures. Overall, our results demonstrate that PAM-distal mutants emerge in liquid cultures expressing PAM-distal mismatched crRNA because of the highly deleterious effect of dual PAM-distal mismatches on Cas12a cleavage.

## Emergence of PAM-distal escape mutants occurs for Cas12a but not Cas9

Our results show that some pairs of PAM-distal mismatches are deleterious enough to cause escape from CRISPR-Cas12a immunity. We also note that Cas12a cleaves in the PAM-distal region of the target [26], and that our and previous results suggest that mutations may arise during repair of DNA breaks induced by Cas endonuclease cleavage [38,39]. Thus, it is possible that Cas12a is uniquely prone to emergence of escape mutations in the PAM-distal region.

To test this hypothesis, we performed similar experiments for *Streptococcus pyogenes* Cas9 (SpCas9), which cleaves targets in the PAM-proximal region [55,65] and therefore may be more prone to emergence of mutations in the PAM and seed region. Similar to FnCas12a, in vitro cleavage assays using SpCas9 revealed that 2 PAM-distal mismatches cause a significantly larger defect than a seed and PAM-distal mismatch (Figs 4A and S9 and S1 Data). Surprisingly, for the target tested in our in vitro cleavage assays, the PAM-distal mutation was more deleterious than the seed mutation even when targeted by the perfect crRNA. These results suggest that PAM-distal mutations should be sufficient to cause escape from SpCas9-mediated immunity.

To test whether such mutants emerge, we performed phage challenge assays in *E. coli* expressing SpCas9 programmed with single-guide RNA (sgRNA) containing mismatches at the same positions relative to the PAM as those tested for FnCas12a (Figs 4B and S10A and S2 Data). Similar to FnCas12a, mismatches caused minimal defects in SpCas9-mediated phage defense on solid media (S10A Fig). In liquid media, delayed lysis occurred in all cultures, including those expressing perfect crRNAs (Fig 4B). To determine whether lysis occurred due to the emergence of phage mutants, we PCR amplified the target regions of phage collected from these lysates and sequenced the amplicons by high-throughput sequencing (S7 Data). Mutants emerged in all samples and target mutations were confined to the PAM and seed, although the positions of these mutations varied (Figs 4C and S10B). Seed and mid-target crRNA mismatches caused a shift away from the PAM and into the seed region. Unlike Cas12a, no PAM-distal mutants emerged for either target when challenged by Cas9 bearing PAM-distal mismatched sgRNAs. These results suggest that both Cas effector specificity and cut site may impact the location within targets at which escape mutations may emerge.

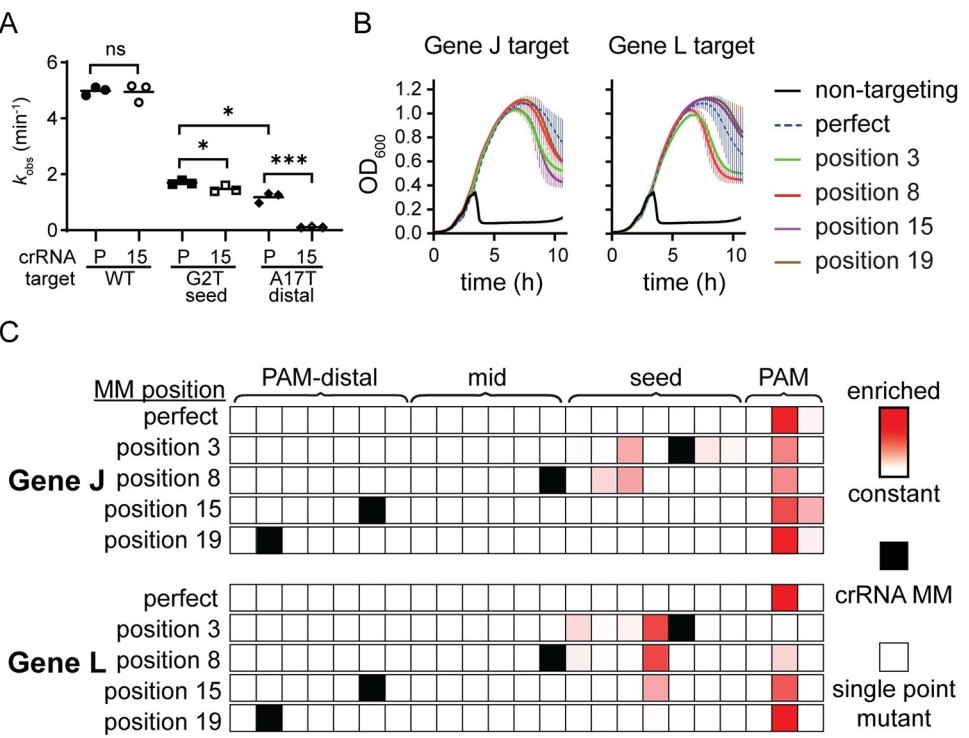

**Fig 4. Cas9 challenge does not cause emergence of PAM-distal mutants.** (A) Observed rate constants for cleavage of a target plasmid bearing a wild type (WT), seed mutant (G2T) and PAM-distal mutant (A17T) gene L target sequence. Cas9 cleavage was measured for both the perfectly matched crRNA (P) or the MM15 crRNA (15). Significance was tested pairwise for all crRNA/target combination by unpaired two-tailed *t* test. * $P \leq 0.05$, *** $P \leq 0.001$, ns $P > 0.05$. Pairwise comparisons for which $P$ value are not indicated had a $P < 0.0001$. See S9 Fig and S1 Data for crRNA and target sequences, gels, and quantified and fit data. (B) Growth curves of *E. coli* expressing Cas9 and sgRNAs bearing either a non-targeting sequence, the perfectly matching spacer sequence (perfect), or a spacer containing mismatch at the indicated position with respect to the PAM. Phage was added when the cells reached mid log phase at approximately 2 h after inoculation. The average of 3 replicates is plotted for each condition, with error bars representing standard deviation. See S2 Data for quantified data. (C) Heat maps showing the location of enriched phage mutations in target regions at the 8 h time point for gene J or L targets after Cas9-mediated selection. Z-scores for abundance of single-nucleotide variants, including nucleotide identity changes or deletions, were determined for each sample relative to the non-targeted control phage population. Enriched sequences indicate high Z-scores. Z-scores range from 0 (white) to 7.0 (darkest red). See S7 Data for quantification of variant abundance.

## Phage targeted with mismatched spacers develop conditional escape mutations

Our results suggest that individual mismatches are often not sufficiently deleterious to allow phages to escape Cas12a targeting. Instead, the combination of the preexisting mismatch and an additional mutation in the target is necessary for escape to occur. It remains unclear to what extent these new mutations contribute to phage escape in the presence of a preexisting mismatch. We chose to pursue further experiments using the crRNA with a seed mismatch targeting gene J because although it was highly deleterious for cleavage in vitro (Fig 1C), it caused rapid phage mutation in liquid culture (Fig 2B). In addition, this mismatch caused the largest variety of mutants to arise for all the crRNAs we tested with mutations at nearly all positions in the seed region (Figs 2C and S2). We hypothesized that this target in an intergenic region was less restrictive of mutation, exacerbating the defect of this crRNA mismatch in vivo.

To test this hypothesis, we generated mutated phage populations using the seed mismatched crRNA targeting gene J. We first infected *E. coli* cells expressing the mismatched crRNA with lambda phage in liquid culture at a wide range of MOIs (Fig 5A). The phages

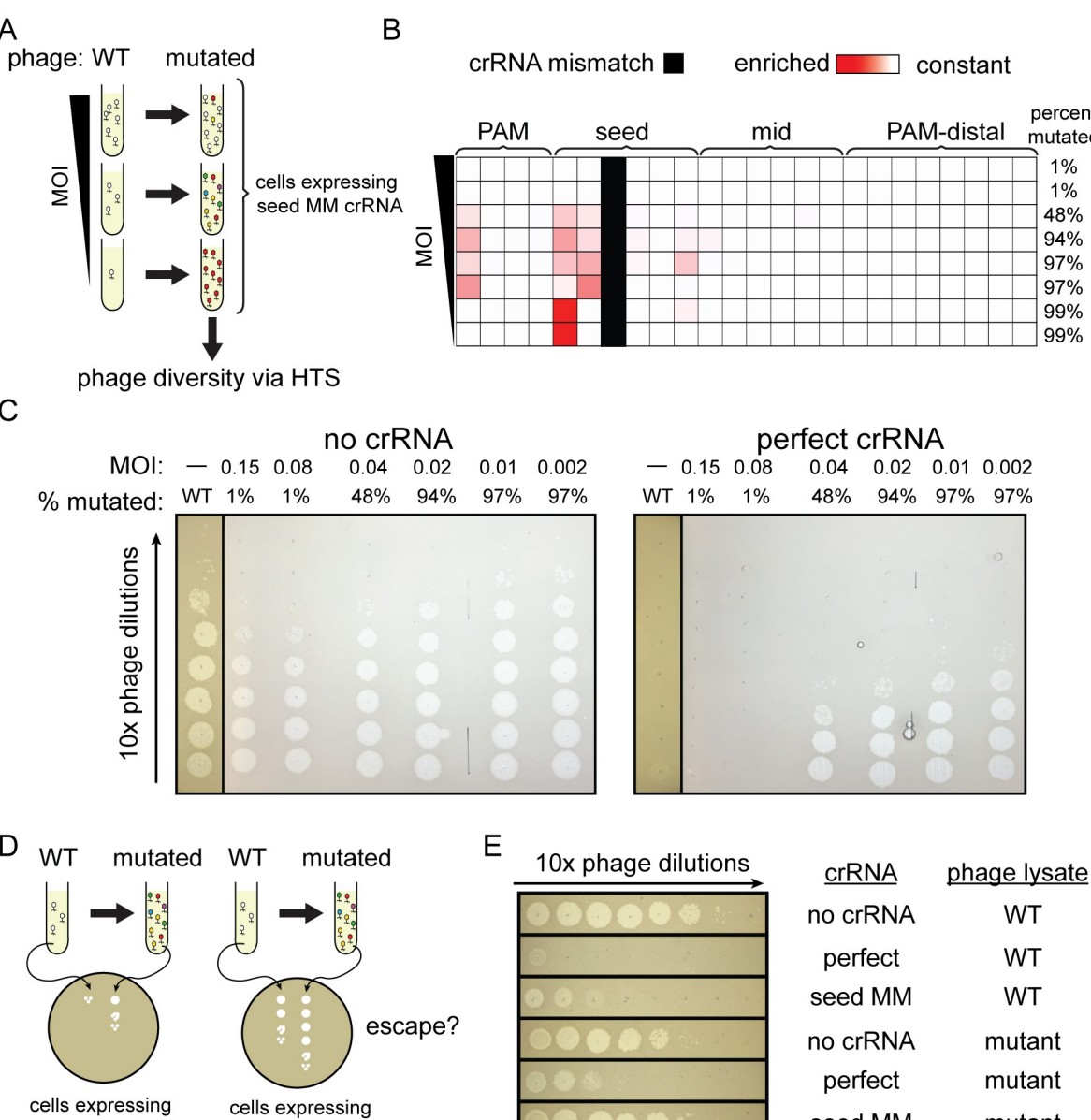

**Fig 5. Combined mismatches are necessary for complete phage escape.** (A) Schematic for experiment to test the impact of MOI on escape phage diversity. Cultures expressing Cas12a and the position 3 mismatched crRNA targeting gene J were infected with lambda phage at varied MOIs. Mutant phages in lysates were detected by high-throughput sequencing. (B) Heat map showing the position of phage mutations that arose when infecting bacteria expressing seed mismatch crRNA at different MOIs. Phage was harvested from liquid cultures containing bacteria expressing FnCas12a and a crRNA with a C-T mismatch at position 3. Phage was added to the culture at mid-log phase at a range of MOIs starting at 0.15 and serial 2-fold dilutions from 1/2 to 1/32 and an additional sample at an MOI of $1.5 \times 10^{-3}$. Phage was harvested 5 h after infection. High-throughput sequencing was used to determine the percent of phages in each that had a mutation in the target region. The heat map shows the positions in the target that were enriched with mutations. These positions are colored darker red according to their Z-score relative to the control phage population. Z-scores range from 0 (white) to 10.1 (darkest red). The position of the initial crRNA mismatch is indicated in solid black. See S8 Data for quantification of variant abundance. (C) Spot assays using phage isolated from liquid cultures as described in (A) on bacteria expressing a matching crRNA. Phage harvested in (A) was 10-fold serial diluted and spotted on bacteria with a crRNA matching the wild-type lambda phage genome target (matching crRNA) or bacteria without a crRNA guiding Cas12a (no crRNA). Wild-type phage controls were spotted on these same bacterial strains. Phages harvested from the lowest MOI cultures were omitted due to their low titer which prevented visible plaque formation on the CRISPR active *E. coli* strain. See S11B Fig for full plates. (D) Schematic for experiment shown in panel (E). Wild-type or mutant phage populations were used for spot assays on plates with lawns of *E. coli* expressing Cas12a and either the perfect or the seed mismatched crRNA to determine whether the combination of the preexisting mismatch and newly acquired target mutations are necessary for complete escape from Cas12a targeting. (E) Spot assays using mutationally diverse phage on bacteria expressing crRNAs with and without mismatches. Phage was isolated from the liquid culture as

described in (A) that was initially infected with phage diluted 1:8. Mutated phage and unmutated control phage (WT) were then used for spot assays on bacterial lawns expressing FnCas12a and a matching crRNA (perfect), a crRNA with the original seed mismatch, or no crRNA as negative control.

were able to clear the culture at MOIs greater than $1.5 \times 10^{-3}$ (S11A Fig and S2 Data). We then analyzed the genomic diversity of the phage population in the targeted region using high-throughput sequencing (S8 Data). The phage population retained the wild-type sequence of the target region at the 2 highest MOIs tested (0.15 and 0.075 MOI), indicating that the wild-type phage can overcome Cas12a-mediated immunity when the bacteria are exposed to enough phage particles (Fig 5B). At the lowest MOIs tested, $1.5 \times 10^{-4}$ and $1.5 \times 10^{-5}$, 99% of the phage population contained a single mutation at the first position of the protospacer. Mutations that arose were most varied at intermediate MOIs. These mutations were in the seed region or mid target region near the existing crRNA mismatch. The number of different mutations observed was also higher compared to the bacterial strain with a matching crRNA to the WT phage target.

Next, we harvested phage from the cultures at all of the MOIs tested and compared protection against this mutant phage population by a crRNA that perfectly matched the wild-type target and a crRNA bearing the original seed mismatch used to generate the mutant population. Visible infection using these new phage lysates was first observed when using phage that was 48% mutated (Figs 5C and S11B). While the perfect crRNA still offered some level of protection against the mutated phage, the crRNA containing the mismatch resulted in complete loss of protection (Fig 5D and 5E). These results indicate that some mutations that emerge in the phage population are only significantly deleterious to Cas12a interference in the presence of the preexisting mismatch, revealing the importance of combined mismatches for phage escape.

## Combining mismatched spacers increases level of protection

Our results indicated that loss of protection by Cas12a due to crRNA mismatches was only partially caused by loss of Cas12a cleavage due to the preexisting mismatch and that mutant emergence generating a second mismatch also contributed substantially to this loss of protection. These results imply that Cas12a mismatch tolerance should enable stronger and longer term protection under conditions where phage mutants are less likely to emerge. To further test this, we introduced both the gene J and L crRNAs into a CRISPR array for co-expression of both crRNAs (Fig 6A). It has been demonstrated previously that bacterial populations with diverse spacer content in their CRISPR arrays show robust immunity against phage at the population level as phage are unable to develop escape mutations in all of their targeted genomic regions [66,67]. Cas12a expressed along with multiple different cRNAs shows increased phage resistance compared to when a single crRNA is present, but the mechanism of this increased resistance is not clear [64]. We investigated this mechanism further in the context of our previous experiments with mismatched crRNAs. If the loss of protection due to a crRNA mismatch is caused only by a slowing of the rate of cleavage, then 2 different mismatched spacers should not provide more protection than 1 spacer repeated twice. Alternatively, if phage mutant emergence significantly contributes to loss of protection in the presence of a crRNA mismatch, 2 different mismatched spacers should provide better protection than a single-mismatched spacer repeated twice.

Consistent with the second possibility, the CRISPR construct with 2 unique mismatched spacers (hereafter referred to as double spacer construct) showed a significantly higher level of protection than either of the crRNA constructs with 2 copies of a single-mismatched spacer

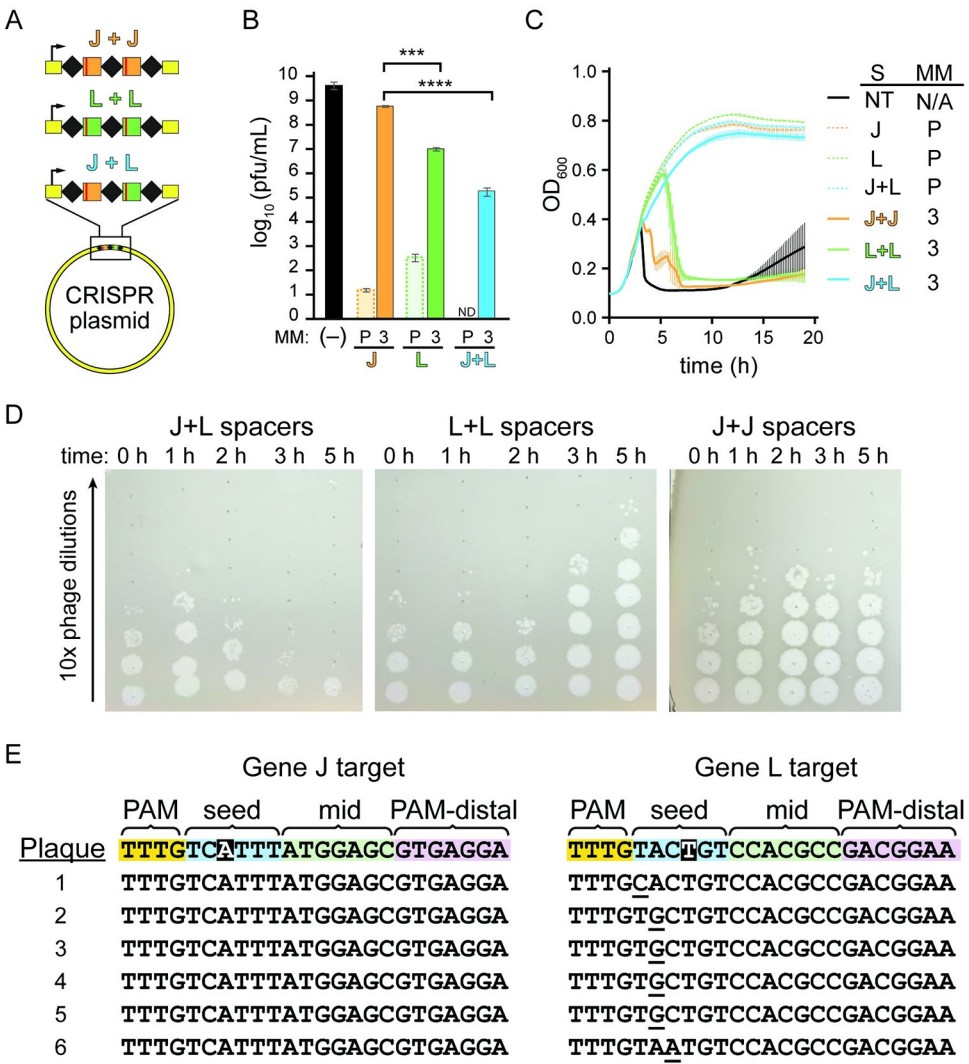

**Fig 6. Multiple mismatched crRNAs provide more protection than individual mismatched crRNAs.** (A) Schematic of experiment in which 2 crRNAs bearing mismatches at position 3 are expressed from a CRISPR plasmid. The CRISPRs either have 2 identical spacers targeting gene J (J+J) or gene L (L+L), or have 2 different spacers targeting gene J and L (J+L). (B) Number of plaques formed on lawns of bacteria expressing multiple mismatched crRNAs. Plaque assays were performed using bacteria containing a plasmid expressing FnCas12a along with different crRNA expression plasmids. Plasmid expressed either the perfect crRNA (P) or the position 3 mismatched crRNA (3). For the perfect crRNAs, plasmids expressed either 1 (gene J or gene L alone) or 2 (J+L) spacers. For the position 3 mismatched crRNAs, each crRNA expression plasmid contains 2 spacers: both targeting gene J both targeting gene L and 1 spacer targeting each of gene J and gene L (J+L). (-) indicates a negative control in which no crRNA was expressed. Plaque forming units (pfu) was calculated using the number of plaques on each plate and the volume of phage lysate added. The average of multiple replicates ($n$ = 2 to 4) is plotted with error bars representing standard deviation. Unpaired, two-tailed $t$ tests were used to determine the statistical significance of each of the single spacer constructs compared to the double spacer construct, *** $P \leq 0.001$, **** $P \leq 0.0001$. See S2 Data for quantified data. (C) Growth curves using the same bacterial strains described in (A). Phage was added when the cells reached mid log phase at approximately 2 h after inoculation. The average of 3 replicates is plotted for each condition, with error bars representing standard deviation. See S2 Data for quantified data. (D) Spot assays used to measure the titer of phage over time in phage infection cultures. Spot assays were performed using 10-fold serial dilutions of phage harvested from cultures in (B) that infected bacterial strains with 2 mismatched spacers at different time points on lawns of CRISPR-inactive *E. coli*. (E) Sequences of both CRISPR targets in single phage plaques for phage harvested from *E. coli* cultures expressing a double spacer construct. The 2 crRNAs contained mismatches at positions highlighted in black. Target regions for the gene J and gene L target were sequenced for 6 individual plaques using Sanger sequencing. Target sequences are aligned to the WT sequence (top row) and mutations are underlined. See S12B Fig for chromatograms.

(hereafter referred to as single spacer construct) when measured by plaque assay (Fig 6B and S2 Data). Similar to liquid cultures with bacteria expressing a single copy of the crRNA, we observed faster lysis of the gene J targeting crRNA in comparison to the gene L targeting crRNA, consistent with the higher chance of escape mutant emergence against the gene J crRNA. Bacteria expressing the double spacer construct showed slowed growth between 1 and 2 h but recovered quickly and did not lyse over the time course tested (Fig 6C and S2 Data). Phage from these cultures was harvested over time and used to infect CRISPR inactive bacteria to determine the relative titers. Phage titers decreased over time in cultures expressing the double spacer construct, while the phage titer increased over time in cultures with cells expressing the single spacer constructs (Fig 6D).

Spotting these same phage lysates on CRISPR active cells showed no noticeable infection by lysate harvested from the double spacer culture, but moderate infection by the single spacer lysate (S12A Fig), suggesting that escape mutants did not emerge from bacteria expressing 2 different mismatched crRNAs. Consistently, sequencing of both target regions in individual plaques revealed mutations in only 1 of the 2 target regions (Figs 6E and S12B). Together with our previous results, these results suggest that loss of protection due to a crRNA mismatch is caused by a combination of loss of Cas12a targeting and the emergence of mutant phages that further block CRISPR interference.

## Preexisting target mutations cause different CRISPR escape outcomes

We have shown that target mismatches artificially introduced by changing crRNA sequences accelerate phage escape and increase the diversity of mutations that appear. We wanted to determine if the same effect would appear if the crRNA-target mismatch was instead caused by a phage genome mutation. We first generated clonal phage populations with single target mutations by isolating individual plaques of mutant phage that emerged following exposure to Cas12a bearing various crRNAs (Fig 7A). Using a crRNA containing a seed mismatch, we isolated phages with mutations in the PAM (T-2C) or seed (C2A) (S13A and S13B Fig), while a crRNA containing a mismatch at position 19 allowed us to isolate 2 separate plaques containing phage with a mutation at position 16 (G16T) (S13C and S13D Fig). After propagating phage from these plaques, we challenged the mutant phages to CRISPR pressure by bacteria expressing crRNAs with a spacer matching the wild-type phage genome target. We observed that the phage with a seed region mutation caused rapid lysis of CRISPR active bacteria (Fig 7B and S2 Data). This indicated that the C2A mutation was a complete escape mutation. However, phage mutations in the PAM or PAM-distal region caused delayed lysis to occur. In particular, of the 2 G16T isolates, only one caused lysis to occur in some of the experimental replicates (Fig 7B and S2 Data).

We harvested phage from the previous cultures and sequenced PCR amplicons of the phage genome targets using Sanger sequencing. In the seed mutant (C2A) phage cultures, the phage retained the same seed mutation and did not develop additional mutations (Figs 7C and S13B), further indicating that C2A is a bona fide escape mutation on its own. In phage with preexisting mutations in the PAM, mutations appeared at the edge of the seed region (Figs 7C and S13A). In phage with a preexisting mutation in the PAM-distal region at position 16, mutations appeared at positions 14 or 18 for phage harvested from cultures that lysed. Repeating the same experiment with PAM-distal mutants in the gene L target similarly caused further mutations to occur in the mid-target and PAM-distal regions (S13E–S13H Fig).

We proceeded with further experiments using only replicates in which a clonal phage population was generated based on an unambiguous Sanger sequencing chromatogram (S13A, S13B, and S13D Fig). Using these phages, we sought to verify that these second mutations

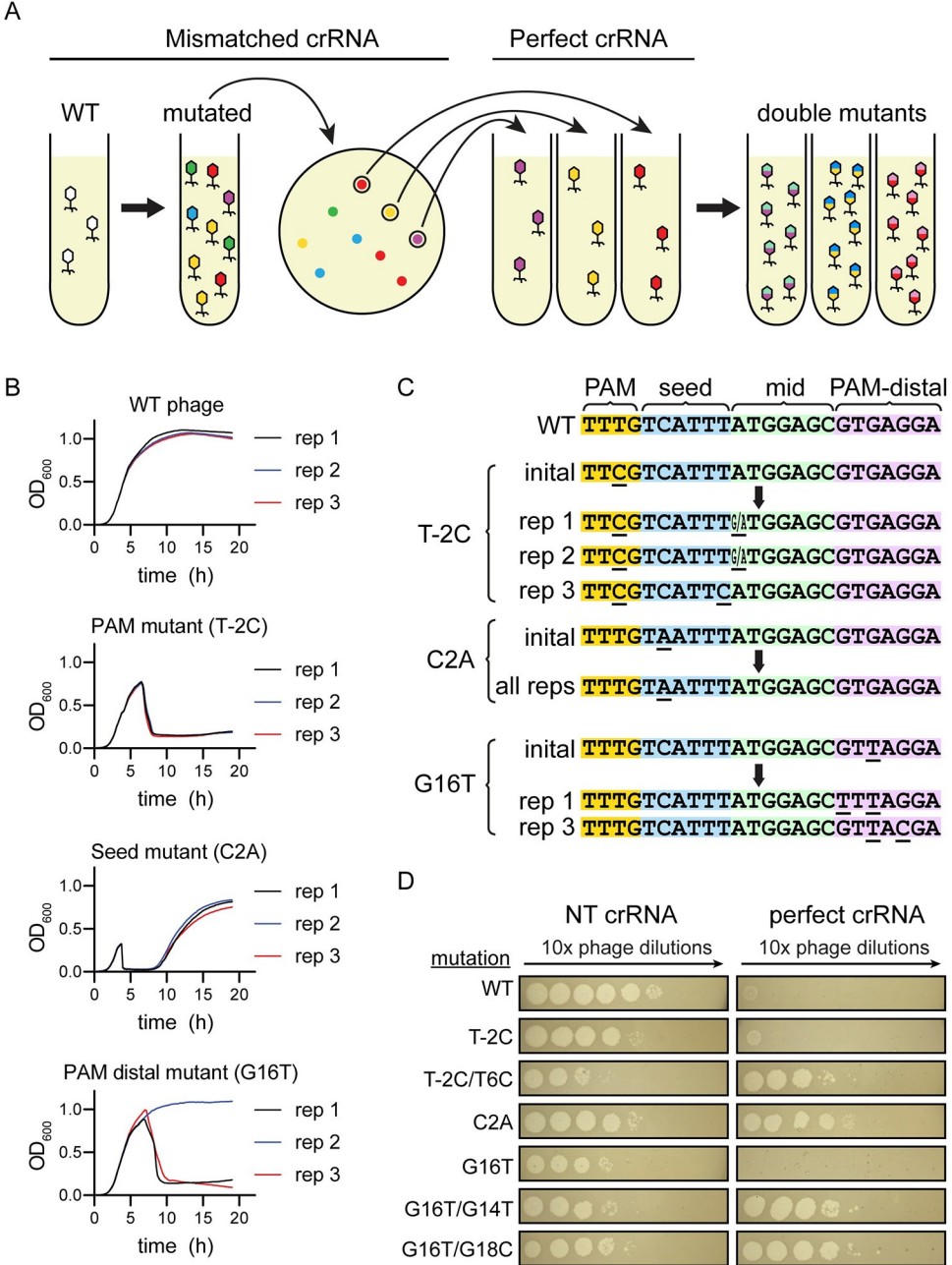

**Fig 7. Generation of double-mutant phage is driven by insufficiently deleterious mutations.** (A) Schematic of the process for generating and purifying single-mutant phage populations. Wild-type phage was used to challenge *E. coli* expressing a crRNA with a mismatch to the target in the phage genome in liquid culture. The resulting phage were isolated and used for a plaque assay on lawns of bacteria expressing the same mismatched crRNA. Single plaques were isolated and again used to challenge bacteria expressing a mismatched crRNA in liquid culture, further purifying and propagating single mutants. Finally, single-mutant phages were used to challenge bacteria expressing a perfectly matching crRNA in liquid culture to determine whether second mutations would appear. (B) Growth curves of bacteria expressing a perfectly matching crRNA challenged with wild-type phage and phage with various single target mutations. Locations of the single mutations in the target are labeled (PAM mutant, seed mutant, and PAM-distal mutant). Position and type of mutations are indicated in parenthesis. Three biological replicates are shown separately for each experimental condition. See S2 Data for quantified data. (C) Diagram of initial and selected mutations that appeared when a single-mutant phage was used to challenge bacteria expressing a perfectly matching crRNA. Initial mutants are single mutants that were generated and purified as described in (A). Sequences below arrows show phage mutants that appeared in different biological replicates (rep 1, 2, or 3) after initial mutant phage lysates were used to infect bacteria expressing a perfectly matching crRNA in liquid culture. Positions with ambiguous base calls are

indicated with 2 bases (X/Y) at that position. Target sequences were interpreted from Sanger sequencing chromatograms (see S13 Fig). (D) Spot assays challenging bacteria expressing a perfectly matching crRNA with various single- and double-mutant phage lysates. WT phage or phages with the indicated target mutations were spotted on bacteria expressing a non-targeting crRNA (left column) and a perfectly matching crRNA (right column).

were allowing CRISPR escape. We compared infection of bacteria expressing the matching crRNA by purified phage containing a single target mutation and phage with 2 target mutations. As expected, phage with the seed target mutation infected bacteria expressing the perfect crRNA at the same level as bacteria expressing a non-targeting crRNA (Fig 7D). Phage with single target mutations in the PAM or PAM-distal region infected bacteria expressing the perfect crRNA at a level close to wild-type phage, while phage with a second mutation infected $10^4$ to $10^5$ times more (Fig 7D). We conclude that target mutations that do not lead to significant CRISPR escape can accelerate the appearance of second mutations that allow complete escape.

## Discussion

In order for Cas12a to be an effective immune effector, it must provide immunity from bacteriophage in diverse conditions. The physical environment controls which bacteria are exposed to which phages and the dispersal of new phage particles after infection and lysis [9,68]. We found that Cas12a overall provided more robust immunity on solid media than in liquid culture. The effect of seed crRNA mismatches using either media correlated with the deleterious effect of the crRNA mismatch on the rate of cleavage in vitro. Mid-target and PAM-distal mismatches, however, showed a much more drastic effect in liquid culture than defects observed in vitro or on solid media when Cas12a was expressed from a strong promoter, causing eventual lysis of the bacterial population, sometimes at a rate similar to seed mismatches. These results strongly indicate that the effect of crRNA mismatches varies depending on the environment where phage exposure may occur.

Although phage can escape CRISPR interference with a single mutation in the seed region [10,35], our results indicate that a seed mutant is sometimes not deleterious enough to allow complete escape. Instead, the mechanism of phage escape occurs through the emergence of mutations that further weaken CRISPR interference when a mismatch is present. This is supported by the rapid emergence of mutant phage we observed even when a highly deleterious seed crRNA mismatch was present. In addition, the number of different mutations that appeared increased when a crRNA mismatch was present and the distribution of these mutations greatly varied depending on the location of the mismatch. These results suggest that mismatches between the crRNA and target decrease phage protection by broadening the range of mutations that allow escape. This mechanism also explains the deleterious effect of mismatches at some positions outside of the seed region on immunity in liquid culture that does not appear during in vitro cleavage.

In natural settings, spacer-target mismatches are caused by phage genome mutation rather than CRISPR spacer mutation that would change the sequence of the crRNA [12,69]. Phages with a single target mutation might not be able to completely escape from certain CRISPR spacers, requiring selection of a second target mutation for full evasion of CRISPR interference [36]. This scenario may become more likely if the seed region, where mutations would normally arise, is located in a critical part of the genome where mutations are highly deleterious. Supporting this, we isolated phage with single mutations in both intergenic (gene J) and coding (gene L) regions that did not cause significant CRISPR escape that then developed second mutations that allowed full escape when exposed to the matching crRNA. In addition, CRISPR

spacers with multiple mismatches become more abundant over time in natural populations [46]. Thus, the presence of mutations may drive further mutation in CRISPR targets over time.

When a PAM-distal crRNA mismatch or a PAM-distal target mutation was present, mutations arose in close proximity to the preexisting mismatch for Cas12a, but not for Cas9. Our in vitro cleavage results suggest that 2 PAM-distal mismatches may be more deleterious than individual seed mismatches or even combinations of seed and PAM-distal mismatches. Thus, phage mutations that result in multiple PAM-distal mismatches are more likely to be selected than PAM or seed mutants when a PAM-distal mismatch already exists.

The enrichment of PAM-distal mutations after exposure to Cas12a and not Cas9 may be influenced by their cut sites, which are in the PAM-distal region and downstream of the target for Cas12a and in the seed region for Cas9 [26,55,65]. Mutations may be more likely to arise around the cleavage site due to DNA repair that occurs after cleavage by Cas12a or Cas9. Notably, many of the crRNAs used in our study caused initial nicking, rather than complete double-strand cleavage, in in vitro cleavage assays (S1B, S7B, S8B, and S9B Figs). Such nicking events may result in recombination or other DNA repair mechanisms that result in alteration of the sequence around the cleavage site. Indeed, there is evidence that phage escape mutation is accelerated by error prone repair after cleavage by Cas effectors and mediated by phage recombinases, especially when a crRNA provides a low level of protection [38,39]. Lambda phage recombination enzymes in particular have been shown to increase the appearance of mutated phage in populations under CRISPR pressure [39]. Our results suggest that some mutants that arose upon Cas12a challenge preexisted in the population, especially for mutants involving single nucleotide or long deletions. However, we also observed strong evidence that mutants arose following Cas effector targeting, including in phage strains lacking Red recombination machinery. We speculate that preexisting mutations may be propagated in the phage population by lambda encoded recombinases, but that error prone repair following Cas effector cleavage is not dependent on Red recombination. Phage with target mutations in their genome that initially survive interference could be used as recombination substrates to pass along that mutation to other phages in the cell. This could account for the rapid increase in mutant phages we observe and the deleterious effect of phage recombination enzyme knockouts [39].

Overall, our results reveal that mismatches throughout the crRNA-target duplex can drastically decrease protection provided by Cas12a. However, there are fundamental differences between our heterologous system and natural CRISPR-Cas systems. Adaptation is an important part of CRISPR immunity. As phage mutate and become immune to targeting by existing spacers, new spacers from the phage genome can be added to the CRISPR array [11,70]. It remains to be investigated how mismatched spacers contribute to acquisition of new spacers in type V systems, especially using a primed mechanism as occurs in type I and type II systems.

## Materials and methods

### Expression plasmid construction

All primers and plasmids used in this study are listed in S1 Table. Cas12a and Cas9 expression plasmids were constructed using pACYCDuet-1. A gene expressing FnCas12a or SpCas9 was inserted downstream of a pBAD promoter in pACYCDuet-1 using Gibson assembly. Promoters were replaced with a previously described low copy P4 promoter [49] using 1 piece Gibson assembly. crRNA expression plasmids were constructed using pUC19. A pBAD promoter was inserted into pUC19 in the multiple cloning site with Gibson assembly. "Round the horn" PCR and ligation was used to add a mini CRISPR array with 1 or 2 spacers downstream of the

pBAD promoter. The same method was used to replace mini CRISPR arrays with Cas9 sgRNA expression constructs.

### Bacterial and phage strains

*E. coli* strain BW25113, a K12 derivative [71] was used for all experiments along with a modified lambda phage strain ($\lambda_{vir}$) [72] that is locked into the lytic lifestyle.

For all CRISPR interference assays, bacteria were transformed with Cas12a and crRNA expression plasmids by heat shock. Transformants were plated on LB plates containing ampicillin at 100 μg/mL and chloramphenicol at 25 μg/mL to select for plasmids pUC19 and pACYCDuet-1, respectively.

The *red* operon deletion strain was generated by using 20 μL of 1:10 diluted $\lambda_{vir}$ stock to infect *E. coli* BW25113 cells harboring pUC19 plasmid with approximately 800 bp section of the lambda genome inserted in the multiple cloning site with the lambda *red* operon removed. The culture was grown in 2 mL LB media containing 25 μg/mL ampicillin and 10 mM MgSO$_4$. After lysis, cell debris was removed by centrifugation and supernatant containing phage was isolated. Phages were then passaged twice through cultures containing *E. coli* harboring a pACYCDuet-1 FnCas12a expression plasmid and separate pUC19 plasmid allowing expression of a pre-crRNA with 2 spacers targeting different locations in the lambda *red* operon to select for phage with the operon deleted. Approximately 20 μL of previously isolated phage lysate was added to each subsequent culture. These 2 mL LB media cultures contained 100 μg/mL ampicillin, 25 μg/mL chloramphenicol, 20 mM arabinose, and 10 mM MgSO$_4$. Phage lysates were harvested, and deletions were confirmed with PCR amplification of the lambda red operon flanking region and Sanger sequencing of the PCR product.

### Phage spot assays

Overnight cultures were started using *E. coli* transformed with Cas12a and crRNA expression plasmids in LB media with ampicillin and chloramphenicol for selection. The next day, 20 μL of these overnight cultures were used to inoculate 2 mL cultures in LB media containing 100 μg/mL ampicillin, 25 μg/mL chloramphenicol, 20 mM arabinose, and 10 mM MgSO$_4$. The cultures were grown in a shaking incubator at 200 rpm and 37°C until OD$_{600}$ 0.4 was reached, and 300 μL of cell culture was added to 3 mL 0.7% soft agar containing the same concentrations of ampicillin, chloramphenicol, arabinose, and MgSO$_4$. The cell-soft agar mixture was vortexed for 5 s and spread onto an LB plate containing ampicillin and chloramphenicol. The plate was dried for 5 min. Lambda phage suspended in LB media with 10 mM MgSO$_4$ was serially diluted in 7 steps with a 1:10 dilution each step, and 2 μL of each phage dilution was then spotted on top of the soft agar layer and the plate was dried for 10 min. Plates were incubated overnight at 30°C.

### Liquid culture phage assays and growth curves

Overnight cultures were started using a single colony of *E. coli* with Cas12a and crRNA expression plasmids in LB media with ampicillin and chloramphenicol added for selection. The next day, these overnight cultures were used to inoculate cultures 1 to 100 in LB media containing 100 μg/mL ampicillin, 25 μg/mL chloramphenicol, 20 mM arabinose, and 10 mM MgSO$_4$. The cultures were grown in a shaking incubator at 200 rpm and 37°C for 1.5 h until OD$_{600}$ approximately 0.25 was reached.

Lambda phage was added at MOI 0.02, or as indicated in figure legends. For growth curves, OD readings were immediately taken after addition of the phage and this value represented the "0 hour" time point. For growth curves shown in Figs 1D, 5A, and S5A, OD was measured

at 600 nm wavelength every 1 h in a WPA Biowave CD8000 Cell Density Meter if growing in culture tubes. For growth curves shown in Figs 6B and S4B, 150 μL cultures were grown in a TECAN infinite M Nano+ 96-well plate reader at 280 rpm and 37°C and OD measurements at 600 nm wavelength were measured every 10 min.

## Phage plaque assays

*E. coli* bacterial cultures were prepared and induced the same way as in the phage spot assays. Approximately 50 μL of phage diluted between 1:1 and 1:$10^6$ in LB media was added to 300 μL of induced cell culture at $OD_{600}$ 0.4. This mixture was then added to 3 mL 0.7% soft agar containing ampicillin, chloramphenicol, arabinose, and $MgSO_4$ as in the phage spot assays, and the mixture was vortexed for 5 s and spread onto an LB plate containing ampicillin and chloramphenicol. The plate was dried for 10 min and left to incubate at 37°C overnight. Plaques were counted the next morning.

## Cas9 and Cas12a expression and purification

Cas9 and Cas12a proteins were expressed in *E. coli* (DE3) cells. LB broth supplemented with 50 μg/mL kanamycin was inoculated with overnight culture of the cells carrying the expression plasmid in 1:100 ratio. Protein expression was induced by adding 0.5 mM IPTG when the culture reached an OD600 of 0.5 to 0.6. The culture was incubated at 18°C overnight (about 16 h) with shaking.

Cas9 and Cas12a were purified by the following protocol, adapted from previous methods with modifications [53,73]. After harvesting, the cell pellets were resuspended in lysis buffer (20 mM Tris-HCl (pH 8.0), 500 mM NaCl, 5 mM imidazole, and 5% glycerol) supplemented with 1 mM PMSF. The resuspended cells were lysed by sonication and the lysate then was centrifuged to remove insoluble material. The clarified supernatant was transferred to a HisPur Ni-NTA resin (Thermo Fisher Scientific) column pre-equilibrated with lysis buffer without disturbing the pellets. The column was washed with 50 column volumes of lysis buffer, then washed again with 50 column volumes of wash buffer (20 mM Tris-HCl (pH 8.0), 500 mM NaCl, 15 mM imidazole, and 5% glycerol). Elution buffer (20 mM Tris-HCl (pH 8.0), 500 mM NaCl, 250 mM imidazole, and 5% glycerol) was applied to elute bound protein that was collected and cleaved with TEV protease in a 1:100 (w/w) ratio to remove MBP. TEV cleavage reaction was incubated overnight at 4°C during dialysis in dialysis buffer (10 mM HEPES-KOH (pH 7.5), 200 mM KCl, 1 mM DTT, and 5% glycerol). The protein was concentrated and then diluted with dilution buffer (20 mM HEPES-KOH (pH 7.5) and 5% glycerol) to a final concentration of 100 mM KCl. The protein was loaded on a HiTrap Heparin HP (GE Healthcare) column pre-equilibrated with Buffer A (20 mM HEPES-KOH (pH 7.5), 100 mM KCl, and 5% glycerol). The column was washed with 20% of Buffer B (20 mM HEPES-KOH (pH 7.5), 1 M KCl, and 5% glycerol). The protein was eluted with Buffer B by applying a gradient from 20% to 100% over a total volume of 60 mL. Peak fractions were collected and analyzed by SDS-PAGE. Fractions containing interested protein were combined and concentrated to 1 mL volume. The upper concentrator chamber was refilled with SEC buffer (20 mM HEPES-KOH (pH 7.5), 200 mM KCl, and 1 mM DTT) and then centrifuged to 1 mL volume (repeated this step 3 times) in order to exchange buffer. Finally, the concentrated proteins were aliquoted, flash-frozen in liquid nitrogen, and stored at −80°C.

## crRNA and tracrRNA preparation

All crRNAs were in vitro transcribed using short oligonucleotides (IDT) consisting of a T7 promoter region and a template sequence. The oligonucleotides first pre-annealed T7

promoter sequence at 90°C for 2 min and then incubated a room temperate for 10 min. The 500 μL transcription reaction was performed in transcription buffer (40 mM Tris (pH 8.0), 38 mM MgCl2, 1 mM Spermidine (pH 8.0), 0.01% Triton X-100, 5 mM ATP, 5 mM CTP, 5 mM GTP, 5 mM UTP, and 5 mM DTT) with 0.5 μM pre-annealed DNA and excess of T7 RNA polymerase at 37°C for 4 h, and 2X RNA dye was added into the reaction, heated for 5 min at 95°C and then kept on ice. The reaction was run on 10% denaturing acrylamide gel for 3 h with 1× TBE buffer. The crRNA band was visualized under UV-light and was excised from the gel. The gel was crushed and soaked overnight in 1 mL nuclease-free water at 4°C with rocking. The gel tube was centrifuged 5 min at 2,000 X g and the supernatant was transferred to Costar Spin-X centrifuge tube filters (Sigma Aldrich). The tube filter was centrifuged at highest speed for 2 min to collect crRNA solution at the collection chamber. Ethanol precipitation was performed to concentrate crRNA.

tracrRNA were also in vitro synthesized as described above, however, the tracrRNA template was cloned into pUC19 plasmid with an EcoRI restriction site at the end of the template sequence. The tracrRNA plasmid was first linearized with EcoRI and then used as template for in vitro transcription without pre-annealing step.

### In vitro cleavage assays

Cleavage assays were prepared in reaction buffer (20 mM HEPES (pH 7.5), 100 mM KCl, 1 mM $MgCl_2$, 1 mM DTT, and 5% glycerol) with a final concentration of 50 nM FnCas12a or SpCas9. FnCas12a RNP complex was formed by incubating FnCas12a and crRNA at a 1:1.5 ratio at 37°C for 10 min. Cas9 RNP complex was formed by incubating Cas9:crRNA:tracrRNA at a 1:1.5:1.5 ratio at 37°C for 10 min. To initiate cleavage, target plasmid was added to a final concentration of 15 ng/μL and a final reaction volume of 100 μL. The reaction was incubated at 37°C. Aliquots (10 μL) were quenched at 7, 15, 30, 60, 300, 900, and 1,800 s by adding 10 μL phenol-chloroform-isoamyl alcohol (25:24:1 v/v, Invitrogen). After phenol-chloroform extraction, DNA products were separated by electrophoresis on a 1% agarose gel and visualized with SYBR Safe (Invitrogen) staining. The densitometry of individual DNA bands was measured in ImageJ (https://imagej.nih.gov/ij/). The fraction cleaved was determined by dividing the total cleaved DNA (nicked and linearized DNA) by total DNA (nicked, linearized, and supercoiled DNA). Fraction cleaved was plotted versus time and fit to a first-order rate equation to determine an observed rate constant for cleavage ($k_{obs}$). Rates were measured in triplicate.

### High-throughput sequencing sample preparation

Phage samples were isolated from spots in spot assays at the highest phage dilution in which a cleared spot was observed to ensure a diverse population of mutant phages would be sampled. A 1 mL pipette tip was poked through a phage spot and the agar in the tip is suspended in 30 μL deionized $H_2O$ in a 1.5 mL microcentrifuge tube and incubated in a 48°C water bath for 20 min to melt the agar and dissolve the phage particles. The tubes were vortexed briefly and incubated in the water bath for another 10 min. The water/melted agar mixture that contains the phage was removed.

For phage samples from liquid culture, 50 μL of each culture was transferred to a 1.5 mL tube and bacteria were pelleted from the liquid culture by centrifuging at 15,000 rpm for 5 min. The supernatant containing phage particles was then removed.

Approximately 5 μL of phage solution was used as the template for a 25 cycle PCR reaction with primers containing Nextera adapters. These PCR products were cleaned up using the Promega Wizard PCR purification kit and used as a template for an 8 cycle PCR reaction to add barcodes for sample identification. Q5 DNA polymerase (New England Biolabs) was used for

all adapter and barcode PCR reactions. Samples were pooled and gel purified using the Promega Wizard PCR purification kit. Gel purified samples were then submitted for MiSeq high-throughput sequencing. Conditions for MiSeq runs were Nextera DNA MiSEQ 150-Cycle which included two 75 base pair paired end reads. Adapter PCR primers were designed so both of the paired R1 and R2 reads overlapped in the entire protospacer region including the PAM.

Samples were prepared for PacBio sequencing by 35 cycle PCR amplification of phage samples isolated from liquid culture. PCR products were purified using the Promega Wizard PCR purification kit and submitted for PacBio sequencing.

## High-throughput sequencing data processing

A script written in Python 3.8 was used to process fastq data files received from a MiSeq run. The script extracts target region sequences and determines if the target region contains a mutation relative to the wild-type target sequence. Base substitutions and deletions were classified along with the location of the substitution or deletion relative to the PAM sequence of the target. Mutations were also classified based on the type of mutation (A to C for example). A Microsoft Excel sheet was then created with the data using the "Xlsxwriter" Python package [74]. Z-score calculations and heat maps for each sample were created using Microsoft Excel. Z-scores for each position in the phage target regions were calculated using the average proportion of reads with mutations at each position in experimental samples $\underline{x}$ and control samples $\underline{\mu}$ and the standard deviation of the proportion of mutations at all positions in all Samples $\sigma$.

$$Z = \frac{\underline{x} - \underline{\mu}}{\sigma} \tag{1}$$

A separate script written in Python 3.8 was used to process data files received from PacBio high-throughput sequencing and find deletions in the lambda phage genome. Sequences were extracted from fastq files and matched piecewise to the WT sequence of the genome region that was PCR amplified. Deletions are output as coordinates in the PCR amplified region and these coordinates were translated to the lambda phage genome to create the bar graph in Fig 7B.

Nucleotide diversity was calculated using the proportion of all pairs of sequences $x_i$ and $x_j$ in each sample and the number of nucleotide differences between each pair of sequences $\pi_{ij}$ divided by the length of the PAM and protospacer region, which is 24 nt in length.

$$Diversity = \frac{1}{24} \sum_{i=2}^{n} \sum_{j=1}^{i-1} 2 x_i x_j \pi_{ij} \tag{2}$$

Find the scripts at https://github.com/alexsq2/lambda-phage-CRISPR-mutants.

## Generation and purification of mutant phage

The gene L A2T mutant phage reported in Fig 3 was generated on solid media by isolation of single plaques. In this case, 300 µL of *E. coli* BW25113 transformed with the FnCas12a and MM15 crRNA expression plasmids at OD600 approximately 0.4 was added to 3 mL 0.7% agar containing 100 µg/mL ampicillin, 25 µg/mL chloramphenicol, 20 mM arabinose, 10 mM MgSO$_4$, and 20 µL of undiluted WT phage lysate. Soft agar was vortexed for 5 s and poured onto LB agar plates containing the same concentrations of ampicillin and chloramphenicol. Plates were incubated at 37°C overnight. Phage was isolated from a single plaque by poking into the plaque with a 1 mL pipette tip and the agar was suspended in 20 µL deionized H$_2$O in a 1.5 mL microcentrifuge tube and incubated in a 48°C water bath for approximately 20 min. Soft agar containing phage was transferred to a fresh 1.5 mL tube.

For mutants reported in Fig 7, overnight cultures were started using a single colony of *E. coli* with Cas12a and indicated crRNA expression plasmids in LB media with ampicillin and chloramphenicol added for selection. The next day, these overnight cultures were used to inoculate cultures 1 to 100 in LB media containing 100 μg/mL ampicillin, 25 μg/mL chloramphenicol, 20 mM arabinose, and 10 mM $MgSO_4$. The cultures were grown in a shaking incubator at 200 rpm and 37°C for 1.5 h until $OD_{600}$ approximately 0.4 was reached. Lambda phage was added at MOI 0.02. Cultures continued to grow in the shaking incubator for 5 h. Cultures were transferred to 1.5 mL tubes and centrifuged at 5,000 rpm for 5 min. Supernatant containing phage was transferred to a fresh 1.5 mL tube.

In both cases, phage was then diluted and used for phage plaque assays on lawns of bacteria expressing the same crRNA as in the previous infection to select against remaining WT phage. Single plaques were isolated and used to infect bacterial cultures again expressing the same crRNA under the same conditions as described above. Cultures were grown in a shaking incubator at 200 rpm and 37°C for 5 h. Approximately 5 μL of phage solution was then used as a template for a 35 cycle PCR reaction with Phusion polymerase to amplify the target region. Sanger sequencing was used to confirm the presence and purity of mutations in the target region.

### Mutant phage competition assay

Single-mutant phages were purified as in "Generation and purification of mutant phage". Two different mutant phage lysates were mixed in 1:2 and 2:1 ratios by titer and these mixes were serially diluted 1:10 in 7 dilution steps in LB media containing 10 mM $MgSO_4$, and 3 μL of each dilution was added to 150 μL *E. coli* BW25113 cultures at OD600 0.4 with cells harboring a pACYCDuet-1 FnCas12a expression plasmid and separate pUC19 plasmid allowing expression of a pre-crRNA targeting the region of the lambda phage genome containing the mutation. These 150 μL LB media cultures contained 100 μg/mL ampicillin, 25 μg/mL chloramphenicol, 20 mM arabinose, and 10 mM $MgSO_4$. Complete lysis was observed for all cultures at 8 h after infection and phage lysates were isolated by centrifugation and removal of the supernatant. Sequencing samples were prepared as in "High-throughput sequencing sample preparation" to determine the proportion of each mutant phage in each sample. Ratio of seed:PAM-distal mutants in the population were determined by dividing the number of reads for the seed mutant by the number of reads for the PAM-distal mutant for each sample.

### Sanger sequencing of phage plaques or phage lysate

Phage was isolated from a single plaque by poking into the plaque with a 1 mL pipette tip and the agar was suspended in 20 μL deionized $H_2O$ in a 1.5 mL microcentrifuge tube and incubated in a 48°C water bath for approximately 20 min. Melted agar and $H_2O$ mixture containing phages was transferred to a 1.5 mL microcentrifuge tube.

Phage was also isolated from liquid cultures by transferring 1 mL of liquid culture to a 1.5 mL microcentrifuge tube and centrifuging at 15,000 rpm for 5 min. Supernatant containing phages was transferred to a clean 1.5 mL microcentrifuge tube.

Approximately 5 μL of phage solution was used as a template for a PCR reaction that amplifies the target region in the middle of approximately 800 base pair PCR product. One of the primers used for the PCR reaction was used for sequencing of the PCR product.

### Supporting information

**S1 Fig. Cleavage assays by FnCas12a with single mismatch crRNAs.** (A) Sequence of the target DNAs, perfectly matching crRNAs and single-mismatched crRNAs. (B) Representative agarose gels showing time course cleavage of negatively supercoiled plasmid (nSC) using

perfectly matching crRNA or single-mismatched crRNA by FnCas12a, resulting in linear (li) and/or nicked (n) products. Time points at which the samples were collected were 7 s, 15 s, 30 s, 1 min, 2 min, 5 min, 15 min, and 30 min. All controls were performed under the same conditions as the longest time point for the experimental samples. (C) Quantification of cleaved products (linear and nicked fractions) from the time course cleavage. Averages of the cleaved fraction values were plotted versus time and fit to a first-order rate equation with error bars; $n = 3$ replicates. For values reported in Fig 1C, each individual replicate was fit, and $k_{obs}$ was reported as the average value for the 3 replicates. See S1 Data for data quantification. (EPS)

**S2 Fig. Analysis of phage mutations that emerge following exposure to Cas12a-mediated interference with mismatched crRNAs.** (A) Line graphs showing the nucleotide diversity of phage target regions over time after exposure to bacteria cells expressing crRNAs with and without mismatches. Target regions are gene J or gene L and crRNAs either match the target region (perfect) or contain mismatches at position x. Nucleotide diversity is calculated using the proportion of each sequence in the sample and the number of nucleotide differences between each pair of sequences. Two individual replicates are shown for each condition. See S3 Data for quantified data. (B) Heat maps showing location of target mutations that arose due to CRISPR targeting by FnCas12a on a solid medium. Z-scores for abundance of single-nucleotide variants, including nucleotide identity changes or deletions, were determined for each sample relative to the non-targeted control phage population. Enriched sequences indicate high Z-scores. Z-scores range from 0 (white) to 7.7 (darkest red). Single-nucleotide deletions are shown at adjacent position to the 3′ side. Positions with crRNA mismatches are labeled with solid black boxes in the heat map. See S4 Data for variant abundance quantification. (C) Graphs showing single-nucleotide variations for mutated phage target sequences present at the 8 h time point for 2 individual replicates. Bar graph height shows the proportion of sequences in each sample with the mutation type at each position in the target. Deletions (Δ) are plotted at the first position where a mismatch occurs between the crRNA and the target. See S3 Data for variant abundance quantification. (EPS)

**S3 Fig. MiSeq sample counts and R1/R2 file overlap.** (A) Table showing absolute counts from MiSeq for each replicate of the 8 h time point for each experimental condition for *E. coli* infected with wild-type phage. Each count represents an extracted sequence in which R1 and R2 reads matched. The negative control (non-targeting crRNA) samples were run in a separate MiSeq run to maximize the number of reads and minimize barcode overlap with mutated samples, allowing for analysis of preexisting mutants in the wild-type population. (B) Bar charts showing mutated sequences at each position in the high-throughput sequencing reads of the negative control lambda phage population for the gene J and gene L region. The target region is highlighted with a red box. R1 and R2 reads do not overlap in the target region (no overlap) or overlap in the target region (target overlap). R1 reads are used for the target region in the no overlap condition. When R1 and R2 reads overlap, sequences in which the target region sequence does not agree for both the R1 and R2 reads are removed from analysis and are not shown in this figure. This measure was taken to ensure that variations observed in negative control samples arose solely from PCR errors or the natural variation of the population. While some variations were still observed with stringent R1/R2 overlap enforced, it is not possible to distinguish PCR errors from natural variation. No mutations were substantially enriched outside of the target region for any of the samples tested in this study. See S3 Data for R1 and R2 variant read counts. (EPS)

**S4 Fig. Single deletions enriched by CRISPR exposure.** Bar charts showing single-nucleotide deletions from the lambda phage gene J target (A) and gene L target (B) in phage that were exposed to cells expressing a non-targeting crRNA (CRISPR inactive) and cells expressing crRNAs with and without mismatches to the lambda phage gene J target (C). (A, B) Deletions are mapped along the target sequences for all time points and both biological replicates for the negative control samples. (C) CRISPR active samples shown for gene J target that contained deletion sequences that represented more than 1% of all reads. No such deletions were observed in the gene L target. Deletions were observed in the gene L coding region in phage in the wild-type population. The deletions could remain in genomes in the population as these genomes are packaged along with functional structural proteins in successfully infected cells. See S3 Data for quantification of single-nucleotide deletion abundance.
(EPS)

**S5 Fig. Deletions in non-essential genomic regions that are selected following Cas12a targeting preexist in the phage population.** (A) PCR amplification of regions surrounding essential and non-essential genes targeted by Cas12a. Spot assays were performed using *E. coli* expressing crRNAs that match 2 non-essential (nin204 and nin146) and 2 essential regions (gene J and gene L) of the lambda phage genome. Controls were performed with a plasmid not encoding a crRNA. Phages were isolated from the phage spots and target and flanking regions of the phage genome were PCR amplified and run on an agarose gel. (B) Sanger sequencing chromatograms of phage genome deletions in non-essential regions targeted by Cas12a. Non-essential regions in the lambda phage genome were targeted with matching crRNAs on solid media. Phages were isolated and the target regions were PCR amplified. Single bands were gel purified and PCR amplified in a second round. These second PCR products were sequenced and the chromatograms were aligned to the WT lambda phage genome. Homology at each end of the deletions was identified and highlighted in blue. (C) Map of genomic deletions observed by PacBio sequencing of PCR amplicons from phage unexposed to CRISPR targeting. DNA from lambda phage unexposed to CRISPR targeting was used as a template for PCR reactions that amplified the same non-essential regions as in (A). This PCR product was sequenced with PacBio long-read sequencing and the obtained sequences were matched with the wild-type lambda genome sequence to identify any deletions present. These deletions are plotted on the chart relative to their position in the genome. The quantity of each deletion is identified by a color code. Deletions found by Sanger sequencing in CRISPR active samples are indicated with an asterisk (*). See S5 Data for sequences and quantification.
(EPS)

**S6 Fig. Growth curves and mutant emergence for Δ*red* phage infection.** (A) Growth curves of *E. coli* expressing FnCas12a and crRNAs bearing non-targeting, perfectly matching, or mismatched crRNAs infected with Δ*red* phage. Phage was added when the cells reached mid log phase at approximately 2 h after inoculation. The average of 3 replicates is plotted for each condition, with error bars representing standard deviation. Large error bars indicate that not all replicate cultures lysed. See S2 Data for quantified data. (B) Graphs showing mutation type in Δ*red* phage target sequences present at the 8 h time point. Bar graph height shows the proportion of sequences in each sample with the mutation type at each position in the target. Deletions (Δ) and multiple mutations are plotted at the first position where a mismatch occurs between the crRNA and the target. The average of 3 replicates is shown, with error bars representing standard deviation. See S3 Data for variant abundance quantification. (C) Comparison of target sequences of phage isolated from cultures in (A) containing cells expressing a crRNA targeting gene J with a mismatch at position 8. The WT target sequence is underlined. Three

individual replicates are shown and the percent of each mutant sequence in the sample is listed. Mutated positions relative to the WT sequence are highlighted in orange.
(EPS)

**S7 Fig. Cleavage assays by FnCas12a with double mismatch crRNAs.** (A) Sequence of the gene J target DNA, perfectly matching crRNA and double-mismatched crRNAs. (B) Representative agarose gels showing time course cleavage of negatively supercoiled plasmid (nSC) using perfectly matching crRNA or double-mismatched crRNA by FnCas12a, resulting in linear (li) and/or nicked (n) products. Time points at which the samples were collected were 7 s, 15 s, 30 s, 1 min, 2 min, 5 min, 15 min, and 30 min. All controls were performed under the same conditions as the longest time point for the experimental samples. The gel for the perfect crRNA is reproduced from S1B Fig. (C) Quantification of cleaved products from the time course cleavage. The average cleaved fraction was plotted versus time and fit to a first-order rate equation with error bars representing standard deviation; $n$ = 3 replicates. For values reported in Fig 3B, each individual replicate was fit, and $k_{obs}$ was reported as the average value for the 3 replicates. See S1 Data for quantification. Quantification for the perfect crRNA is also shown in S1C Fig.
(EPS)

**S8 Fig. Cleavage assays by FnCas12a of wild-type and mutant target sequences.** (A) Sequences the perfectly matching crRNA, position 15 mismatched crRNA, and 3 gene L target sequences used for cleavage assays. (B) Representative agarose gels showing time course cleavage of negatively supercoiled plasmid (nSC) bearing a wild-type (WT), A2T, or G17T-containing gene L sequence using perfectly matching crRNA or position 15 mismatched crRNA by FnCas12a, resulting in linear (li) and/or nicked (n) products. Time points at which the samples were collected were 7 s, 15 s, 30 s, 1 min, 2 min, 5 min, 15 min, and 30 min. All controls were performed under the same conditions as the longest time point for the experimental samples. The gels for the perfect crRNA and 15 mismatched crRNA cleaving WT target are reproduced from S1B Fig. (C) Quantification of cleaved products from the time course cleavage. The average cleaved fraction was plotted versus time and fit to a first-order rate equation with error bars representing standard deviation; $n$ = 3 replicates. For values reported in Fig 3D, each individual replicate was fit, and $k_{obs}$ was reported as the average value for the 3 replicates. See S1 Data for quantification. Quantifications for perfect and MM15 crRNA cleaving WT target are also shown in S1C Fig.
(EPS)

**S9 Fig. Cleavage assays by SpCas9 of wild-type and mutant target sequences.** (A) Sequences the perfectly matching crRNA, position 15 mismatched crRNA, and 3 gene L target sequences used for cleavage assays. Cleavage was performed using a crRNA-tracrRNA pair. (B) Representative agarose gels showing time course cleavage of negatively supercoiled plasmid (nSC) bearing a wild-type (WT), G2T, or A17T-containing gene L sequence using perfectly matching crRNA or position 15 mismatched crRNA by SpCas9, resulting in linear (li) and/or nicked (n) products. Time points at which the samples were collected were 7 s, 15 s, 30 s, 1 min, 2 min, 5 min, 15 min, and 30 min. All controls were performed under the same conditions as the longest time point for the experimental samples. (C) Quantification of cleaved products from the time course cleavage. The average cleaved fraction was plotted versus time and fit to a first-order rate equation with error bars representing standard deviation; $n$ = 3 replicates. For values reported in Fig 4A, each individual replicate was fit, and $k_{obs}$ was reported as the average value for the 3 replicates. See S1 Data for quantification.
(EPS)

**S10 Fig. Phage protection by and mutant emergence from SpCas9 with sgRNA mismatches.** (A) Spot assays using lambda phage on lawns of bacteria expressing SpCas9 along with sgRNAs with and without mismatches. SgRNAs target gene J or gene L and contain mismatches at position X or match the target (perfect). (B) Graphs showing single-nucleotide variants in phage target sequences present at the 8 h time point following challenge by Cas9 bearing different sgRNAs. Bar graph height shows the proportion of sequences in each sample with the mutation type at each position in the target. Deletions (Δ) are plotted at the first position where a mismatch occurs between the crRNA and the target. The average of 3 replicates is shown, with error bars representing standard deviation. See S7 Data for variant abundance quantification.
(EPS)

**S11 Fig. Mutant emergence at varied MOIs.** (A) Growth curves using cells expressing a crRNA targeting gene J with a mismatch in the seed region and infected with phage at different MOIs. Phage was added at the indicated MOIs when cells reached mid log phase and the $OD_{600}$ of the culture was measured over time. Cultures at lower MOIs did not lyse and are omitted from the graph. See S2 Data for quantified data. (B) Spot assays estimating the titer of phage lysates exposed to interference by Cas12a armed with a seed mismatched crRNA. Full plates from Fig 4B, including lowest MOI samples which produced phages with low titers.
(EPS)

**S12 Fig. Phage targeted by multiple spacers develops mutations in 1 or more targeted regions.** (A) Spot assays performed using lambda phage that previously infected *E. coli* in liquid culture expressing FnCas12a and 2 different crRNAs targeting gene J and L (lysate spacers J+L) or 2 of the same crRNA targeting gene L (lysate spacers L+L) both with mismatches in the seed region. Phage was harvested at different time points of the liquid culture (0, 1, 2, 3, and 5 h after infection). The previous phage lysates were spotted on cells expressing a crRNA that matches the gene L target in the lambda genome (gene L perfect). (B) Sanger sequencing chromatograms showing sequences of target regions of the genome in phage exposed to cells expressing 2 mismatched crRNAs targeting gene J and gene L, respectively. Phage from single plaques was isolated and both target regions were sequenced. Mutated bases are highlighted.
(EPS)

**S13 Fig. Purified single-mutant and double-mutant chromatograms.** (A–H) Sanger sequencing chromatograms of single- and double-mutant phage lysates. Single-mutant phages were generated by exposure to crRNAs with mismatches (MM crRNA) at different positions (position X) and purified as shown in Fig 7A. Mutants were generated in the gene J and gene L CRISPR target. Purified single-mutant phage was used to challenge bacteria expressing a perfect crRNA and target regions were sequenced by Sanger sequencing to determine if second mutations appeared. Both mixed and clonal double-mutant populations were generated after this step. Mutated bases are highlighted.
(EPS)

**S1 Table. Lists of plasmids, primers, and oligonucleotides used in this study.**
(XLSX)

**S1 Data. Quantification of percent cleaved over time for cleavage assays.** Contains data presented in Figs 2C, 3B, 3D, 4A, S1C, S7C, S8C, and S9C.
(XLSX)

**S2 Data. Quantification of bacterial liquid growth curves and plaque assays.** Contains data presented in Figs 1D, 4B, 6B, 6C, 7B, S6A, and S11A.
(XLSX)

**S3 Data. Quantification of high-throughput sequencing data for gene J and L targets in wild-type and Δ*red* phage upon Cas12a challenge with perfect and mismatched crRNAs in liquid media.** Contains data presented in Figs 2B, 2C, S2A, S2C, S3, S4, and S6B.
(XLSX)

**S4 Data. Quantification of high-throughput sequencing data for gene J and L targets upon Cas12a challenge with perfect and mismatched crRNAs on solid media.** Contains data presented in S2B Fig.
(XLSX)

**S5 Data. Quantification of PacBio sequencing of nin146 and nin204 genomic regions following Cas12a challenge.** Contains data presented in S5C Fig.
(XLSX)

**S6 Data. Quantification of phage competition between phages containing mutations in the seed or PAM-distal region of the gene L target.** Contains data presented in Fig 3F.
(XLSX)

**S7 Data. Quantification of high-throughput sequencing data for gene J and L targets upon Cas9 challenge with perfect and mismatched crRNAs in liquid media.** Contains data presented in Figs 4C and S10B.
(XLSX)

**S8 Data. Quantification of high-throughput sequencing data for gene J target upon phage infection at different MOIs.** Contains data presented in Fig 5B.
(XLSX)

**S1 Raw Images. PDF file containing all raw gel images.** Original gel images for all images presented in the Supporting information figures or used for triplicate quantification of Cas12a or Cas9 cleavage rates. All gels were visualized using SyberSafe staining. When relevant, annotations list which samples were used for gel images in S1B, S5A, S7B, S8B or S9B Figs. Samples that were not included in the Supporting information figures contain replicates that were used for quantification of observed rate constants. All samples are ordered as labeled in the relevant Supporting information figure image.
(PDF)

## Acknowledgments

We thank Michael Baker and Kevin Cavallin of the Iowa State DNA Facility for advice on MiSeq sample preparation and data processing. MiSeq sequencing was performed at the Iowa State DNA Facility and PacBio sequencing was performed by the DNA Sequencing Center of Brigham Young University.

## Author Contributions

**Conceptualization:** Michael A. Schelling, Dipali G. Sashital.

**Data curation:** Michael A. Schelling.

**Formal analysis:** Michael A. Schelling, Giang T. Nguyen.

**Funding acquisition:** Dipali G. Sashital.

**Investigation:** Michael A. Schelling, Giang T. Nguyen.

**Methodology:** Michael A. Schelling, Dipali G. Sashital.

**Project administration:** Dipali G. Sashital.

**Software:** Michael A. Schelling.

**Supervision:** Dipali G. Sashital.

**Validation:** Michael A. Schelling.

**Visualization:** Michael A. Schelling, Dipali G. Sashital.

**Writing – original draft:** Michael A. Schelling, Giang T. Nguyen, Dipali G. Sashital.

**Writing – review & editing:** Michael A. Schelling, Dipali G. Sashital.

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
