## [Editor Report · Decision Letter 0]

18 Jul 2022

Dear Dr. Sashital, 

Thank you for submitting your manuscript entitled "CRISPR-Cas effector specificity and target mismatches determine phage escape outcomes" for consideration as a Research Article by PLOS Biology. I apologize for not sending this email earlier, due to the summer holiday season we are a bit slower than usual. 

Your manuscript has now been evaluated by the PLOS Biology editorial staff and I am writing to let you know that we would like to send your submission out for external peer review.

Once your full submission is complete, your paper will undergo a series of checks in preparation for peer review. After your manuscript has passed the checks it will be sent out for review. To provide the metadata for your submission, please Login to Editorial Manager (https://www.editorialmanager.com/pbiology) within two working days, i.e. by Jul 20 2022 11:59PM.

Kind regards,

Paula

Senior Editor

PLOS Biology

---

## [Decision Letter · Decision Letter 1]

20 Sep 2022

Dear Dr. Sashital,

Thank you for your patience while your manuscript "CRISPR-Cas effector specificity and target mismatches determine phage escape outcomes" was peer-reviewed at PLOS Biology. It has now been evaluated by the PLOS Biology editors, an Academic Editor with relevant expertise, and by several independent reviewers. 

In light of the reviews, which you will find at the end of this email, we would like to invite you to revise the work to thoroughly address the reviewers' reports.

As you will see below, the reviewers agree that a major weakness of the manuscript is the lack of mechanistic insight. This is highlighted by the comments of reviewer #2 and #3. Addressing the reviewers' criticisms would require additional experimental work, improved clarity and depth in your discussion. Please address all the reviewers' comments. 

Given the extent of revision needed, we cannot make a decision about publication until we have seen the revised manuscript and your response to the reviewers' comments. Your revised manuscript is likely to be sent for further evaluation by all or a subset of the reviewers.

**IMPORTANT - SUBMITTING YOUR REVISION**

*Re-submission Checklist*

*Published Peer Review*

*PLOS Data Policy*

*Blot and Gel Data Policy*

Sincerely,

Paula

---

Senior Editor

PLOS Biology

REVIEWS:

Reviewer #1: CRISPR.

Reviewer #2: Phages-host interactions.

Reviewer #3: Phages and CRISPR.

Reviewer #1: CRISPR-Cas systems use RNA-guided nucleases to interfere against foreign genetic elements, like bacteriophage, in a sequence-specific manner. Because targeting is based on homology between the crRNA and the target sequence, phages frequently escape immunity by evolving mutations in the target site. The impact of these mutations on Cas protein recognition and cleavage depends on their location within the target sequence, PAM and seed mutations confer the greatest resistance to CRISPR interference, while PAM-distal mutations usually have minimal effects. Here the Sashital group discovered that mismatches in all areas of the spacer/target restore the ability of phages to overcome interference by the nuclease Cas12a, even if they do not affect target DNA cleavage in vivo or in vitro. The authors show that these mismatches promote the rise of phage mutants containing additional mismatches in the target region, which more completely evade CRISPR immunity. This is a very thorough paper, with nicely presented data that fully supports the conclusions. I commend the authors for their nice work, and support publication with very minor editorial revisions regarding the points below.

The data in Fig 2C show that the additional phage escape mutations that arise in the context of pre-existing crRNA:target mismatches correlate nicely with the site of the pre-existing mismatch. E.g. seed mismatches promote the accumulation of additional seed and PAM mutations, while PAM-distal mismatches accumulate additional PAM-distal mismatches. To me the observation that the observed escape mutations cluster near the prior mismatch is fascinating, and warrants a bit of additional discussion:

1. Does it not suggest that the additional escape mutations are not pre-existing in the phage population and might be instead introduced during repair after Cas12 targeting? Since PAM and seed mutations are strongly advantageous for the phage, and are sufficient to evade targeting on their own, I would expect these to accrue in all circumstances regardless of the location of the previous mismatch, but this is not what is observed. To me the data support a model in which these escape mutations are actively being introduced.

2. Cas12 is known to cleave ssDNA nonspecifically in response to target binding. Could collateral cleavage at the exposed ssDNA at the mismatch bubble be responsible for the accumulation of escape mutations?

In the mutation measurements by deep sequencing in Fig 2, it would be interesting to see the number of mutations in the rest of the phage genome (outside the target region). Could the authors present this data if they have it? 

Line 96: Could the authors specify whether Cas12a was expressed from a plasmid or in the E. coli chromosome, and which promoter was used?

Line 115: Correlation between cleavage of mismatched target DNA and efficacy in EOP assays - There seems to be a discrepancy between the EOP assay and cleavage rate in the gene J target for some targets. In particular, the MM8 target is cleaved as well as P in vitro (and better than MM19), but in vivo MM8 is severely impaired in targeting and MM19 is just as susceptible as P. 

Line 351: "perfect crRNA"

Reviewer #2: Schelling et al investigate the impacts of various spacer mismatches on the efficiency of phage targeting by a Type V-A CRISPR-Cas system and the corresponding phage mutations that arise in escapers at the targeted locus. The researchers use Cas12a from Francisella novicida (FnCas12a) expressed in E. coli and phage lambda as a model host-virus system. They demonstrate that mismatches between two representative spacers and their corresponding protospacers in phage lambda cause defects in immunity in a position-specific manner, with seed region mismatches being most detrimental to CRISPR immunity (Fig. 1). The authors go on to characterize phages remaining in culture after Cas12a challenge and show that the position of protospacer mutations in these 'escapers' differ depending upon the position of the mismatched nucleotide (Fig. 2), and that multiple PAM distal mutations can cause loss of immunity by Cas12a (Fig. 3). The multiplicity of infection (MOI) impacts the diversity of phage escaper mutations in the presence of a single mismatch, and combined mismatches are necessary to promote phage escape (Fig. 4). However, two different crRNAs containing mismatches are more effective against phage than individual mismatched crRNAs (Fig. 5). The authors go on to test the impact of protospacer mutations on the diversity of subsequent mutations by isolating individual phage escapers and challenging those with the WT crRNA (Fig. 6). The authors observed that a single mutation naturally-acquired in the seed region is sufficient to afford full protection, while PAM and PAM-distal mutations require the acquisition of additional mutations to allow for phage escape (Fig. 6). Finally, the authors examine the pre-existing mutations in the phage population and found deletions in non-essential genomic loci which get selected upon Cas12a targeting (Fig. 7). 

Overall, the manuscript is well-written and reports on interesting observations relating to the evolutionary dynamics of a phage population when faced with 'imperfect' targeting by Cas12a. The effects of Cas12a on phage evolution is understudied; therefore, these findings are expected to advance our understanding of the biology and impacts of Type V CRISPR-Cas systems. While some of the findings are not surprising (such as the disproportional impact of seed region mutations on Cas12a function, and the presence of pre-existing deletions in nonessential genomic loci in the phage population), there are some compelling observations relating to the compounded inhibitory effect of multiple PAM-distal mutations on Cas12a function, and the diverse mutations that can arise in the presence of mismatches which vary according to the position of the mismatch. However, one major weakness in the study is the absence of definitive evidence explaining how crRNA mismatch position drives mutations in various locations in the protospacer. This and other specific points are listed below in order of importance. 

1. Figure 7: The conclusion that the SNPs in escaper phages following Cas12a 'imperfect' targeting are pre-existing in the population (lines 403-406) is not convincing. The entire paper is focused on the acquisition of SNPs in the essential genes L and J, but when the authors look for pre-existing mutations in the phage population, they only look for deletions in non-essential loci. Is it possible that their origins differ? Recent studies have shown that DNA cleavage by Types I and II CRISPR-Cas systems drive mutagenesis in Lambda phage, and this is mainly driven by the phage-encoded repair machinery (Hossain et al, 2021). The conclusion would be more convincing if the authors repeated one of the targeting/mismatch assays in using a mutant phage variant lacking repair machinery and show that the frequency and pattern of mutations in escaper phages remains unchanged. 

2. Figure 3C: The authors characterize Cas9 phage escapers in the presence of mismatched sgRNAs and were unable to identify PAM-distal mutations using Sanger sequencing. From this observation, the authors conclude that Cas12a may be unique in its susceptibility to PAM-distal mutations (lines 232-237). However, Sanger sequencing is far less sensitive than Illimuna (which was used for the Cas12a escaper characterization). Is it possible that PAM-distal mutants are present following Cas9 targeting, but rare? Can the authors follow-up with Illumina sequencing of Cas9 escapers to show a definitive difference between the two CRISPR types?

3. Figure 1: Early in the manuscript, the authors should provide more information about genes K and J. What do they encode? Are they essential for phage survival? The latter is discussed somewhere in the middle of the manuscript, but this information should be introduced along with the assay at the beginning since this greatly affects the types of mutations that can be acquired. Also, why were these specific genes chosen as representatives? Do the authors think that the positional effects of mismatches are generalizable for other spacer sequences when targeting essential genes?

4. Figure 2C and D: It is unclear whether the data represents one phage with several point mutations, or many phages, each carrying a smaller subset of the mutations. Can the authors provide more clarification to help guide the reader in the interpretation? 

5. Figure 5 A, B: The perfect match crRNA control is missing.

Reviewer #3: Experiments described in this manuscript investigate the effects of CRISPR RNA (crRNA) mismatches on the ability of phages to escape targeting by the Cas12a CRISPR-Cas system of Francisella novicida. 

The experimental set-up used here involves heterologous expression of Cas12a and crRNA in E. coli. The crRNA molecules expressed were designed to target phage lambda at two different positions. Experiments involved designing spacers with mismatches at several different positions across the spacer region. Many interesting observations arise from these experiments. In particular, the authors find that even though a mismatch has no effect on the ability of the Cas12 complex to cleave DNA in vitro, phages escaping CRISPR immunity arise much more readily when any mismatch is present. Escaper mutations in phages tend to be close to the site of the mismatch in the crRNA. Interestingly, a similar system testing Cas9 gave a very different result. 

The experiments performed here are done rigorously and are clearly described. The results are definitely interesting and warrant publication. My problem with this manuscript with respect to publication in PLoS Biology is that I'm uncertain of the big picture conclusions emerging from this work. While the results are interesting and somewhat unexpected in some instances, it is difficult to draw mechanistic insight from these experiments. Since Cas9 behaved differently, I wonder what the difference is between the Cas12 and Cas9 mechanisms that could cause this. Is it possible that mismatches could affect the Cas12 non-specific nuclease activity that is induced upon target-binding? Why did the authors focus on Cas12? The authors should attempt to draw some more general conclusions from this work if it is possible. 

Some minor comments:

Line 107-Authors should briefly discuss the effects of the point mutations in vitro. Are these the expected results. A reader who is unfamiliar with the system may be expecting larger effects.

Line 169-Authors need to refer to a figure here. 

Figure 4 could be improved by providing schematics of the experiments. These experiments were somewhat complicated.

---

## [Decision Letter · Decision Letter 2]

22 Feb 2023

Dear Dr. Sashital,

Thank you for your patience while we considered your revised manuscript "CRISPR-Cas effector specificity and cleavage site determine phage escape outcomes" for publication as a Research Article at PLOS Biology. This revised version of your manuscript has been evaluated by the PLOS Biology editors, the Academic Editor, and the original reviewers.

Based on the reviews, we are likely to accept this manuscript for publication, provided you address the following data and other policy-related requests.

1. DATA POLICY:

A) Supplementary files (e.g., excel). Please ensure that all data files are uploaded as 'Supporting Information' and are invariably referred to (in the manuscript, figure legends, and the Description field when uploading your files) using the following format verbatim: S1 Data, S2 Data, etc. Multiple panels of a single or even several figures can be included as multiple sheets in one excel file that is saved using exactly the following convention: S1_Data.xlsx (using an underscore).

B) Deposition in a publicly available repository. Please also provide the accession code or a reviewer link so that we may view your data before publication.

Regardless of the method selected, please ensure that you provide the individual numerical values that underlie the summary data displayed in the following figure panels as they are essential for readers to assess your analysis and to reproduce it: Figures 1CD, 2BC, 3BDF, 4ABC, 5B, 6BC, 7B, S1C, and Supplementary Figures S2ABC, S3B, S4, S6AB, S7C, S8C, S9C, S10B, S11A.

**Please also ensure that figure legends in your manuscript include information on where the underlying data can be found, and ensure your supplemental data file/s has a legend.**

We require the original, uncropped and minimally adjusted images supporting all blot and gel results reported in an article's figures or Supporting Information files. We will require these files before a manuscript can be accepted so please prepare and upload them now. Please provide this for Supplementary Figures S5A, S7B, S8B, S9B.

Please carefully read our guidelines for how to prepare and upload this data: https://journals.plos.org/plosbiology/s/figures#loc-blot-and-gel-reporting-requirements

We expect to receive your revised manuscript within two weeks.

*Published Peer Review History*

*Press*

Sincerely,

Paula

---

Senior Editor,

pjaureguionieva@plos.org,

PLOS Biology

Reviewer remarks:

Reviewer #1: The additional experiments and textual changes made by the authors have substantially improved the study, I have no further queries and support publication.

Reviewer #2: The authors have done an excellent job addressing the reviewers' concerns through added experiments and discussion to better support their claims. I see no reason to further delay publication of this manuscript.

Reviewer #3: The authors have done a thorough job of addressing my concerns and those of the other reviewers. I believe that the paper is now suitable for publishing in PLoS Biology.

---

## [Editor Report · Decision Letter 3]

6 Mar 2023

Dear Dr Sashital,

Thank you for the submission of your revised Research Article "CRISPR-Cas effector specificity and cleavage site determine phage escape outcomes" for publication in PLOS Biology. On behalf of my colleagues and the Academic Editor, Jeremy Barr, I am pleased to say that we can in principle accept your manuscript for publication, provided you address any remaining formatting and reporting issues. These will be detailed in an email you should receive within 2-3 business days from our colleagues in the journal operations team; no action is required from you until then. Please note that we will not be able to formally accept your manuscript and schedule it for publication until you have completed any requested changes.

PRESS

Sincerely, 

Paula

---

Senior Editor

PLOS Biology
